# Analysis of influencing factors on review efficiency of multidisciplinary scientific research projects using DEMATEL with a 5-point scale

Junyi Li, Yana Xiao * 

Guangdong Science & Technology Infrastructure Center, Guangzhou, Guangdong, China

* ynaxiao@163.com

## Abstract

In the wake of advancing technology and the convergence of diverse disciplines, collaborative research across academic sectors has become instrumental in fostering innovation and tackling multifaceted challenges. The inherent complexity of such multidisciplinary endeavors, characterized by a myriad of research trajectories and a spectrum of expertise, poses significant challenges to effective review. This study aims to identify and analyze the factors influencing the review efficiency of multidisciplinary scientific research projects to ensure their smooth development and improved quality. To address this challenge, we employ the Decision Making Trial and Evaluation Laboratory (DEMATEL) method with a 5-point scale. First, we introduce an indexing framework to systematically identify factors influencing the appraisal of multidisciplinary efforts. This framework is then complemented by expert-driven questionnaires, harnessing domain-specific insights to ascertain the significance and interconnectedness of these factors. Using the DEMATEL method, we distill the data to identify key influencing factors that enhance review efficiency for multidisciplinary projects. Our findings provide pragmatic strategies and policy guidance, equipping institutional bodies and program leads with tools to refine the review process of multidisciplinary scientific research projects.

## 1 Introduction

In the realm of scientific inquiry, multidisciplinary research is increasingly recognized as an essential approach for addressing complex challenges that transcend the boundaries of individual disciplines [1]. This is particularly relevant in tackling contemporary issues such as climate dynamics, healthcare systems, and technological advancements, which are inherently multifaceted and resistant to unidisciplinary interpretations [2–4]. Such challenges, often resistant to unidisciplinary interpretations, necessitate a synthesis of expertise spanning multiple fields. This integrative approach yields solutions that are both in-depth and encompassing.

**Data Availability Statement:** A minimal dataset of the results described in our manuscript has been uploaded to the Interuniversity Consortium for Political and Social Research (ICPSR), the project

link is https://www.openicpsr.org/openicpsr/project/197681/version/V2/view.

**Funding:** The authors were funded by Guangdong Provincial Science and Technology Plan Project grant number 2020B1010010005. The funders had no role in study design, data collection and analysis, decision to publish, or preparation of the manuscript.

**Competing interests:** The authors have declared that no competing interests exist.

The melding of diverse insights and methodologies heralds the potential for pioneering discoveries, catalyzing the pursuit of innovative concepts that might otherwise remain obscured. This collaborative paradigm not only amplifies the analytical scope, allowing for the discernment of subtle interconnections, but also bolsters the efficacy of problem-solving strategies [5]. A notable attribute of multidisciplinary research is its propensity to bridge disciplinary divides, ensuring fluid dialogue and methodological cohesion, which in turn cultivates a holistic comprehension of diverse viewpoints [6]. Given the pressing imperatives of our times, such as sustainable progression and technological advancements, a multidisciplinary approach becomes indispensable [7]. By harnessing collective expertise, it is poised to deliver robust and enduring solutions. Fundamentally, as this mode of research delves into previously unexplored domains and amalgamates knowledge from a spectrum of disciplines, it often paves the way for the emergence of innovative theoretical constructs, pushing the boundaries of collective understanding [8].

Recent developments in digital innovation and the formation of innovation networks, particularly in green and sustainable technologies, underscore the need for effective multidisciplinary collaboration [9–11]. These innovations require robust frameworks for evaluation and implementation, making the efficiency of the review process critically important. Nations globally have also recognized the imperative of multidisciplinary research in addressing complex, multifaceted challenges that single disciplines might find challenging to tackle in isolation [12]. For instance, in the realm of big data research, countries like the United States and China have been at the forefront, channeling significant resources to foster interdisciplinary collaborations [13]. Such endeavors are not merely limited to financial allocations but extend to creating conducive ecosystems that nurture the confluence of varied disciplines [14]. Digital technology plays a crucial role in the industrial structure upgrading process. It facilitates the integration of advanced technologies, enhances productivity, and drives innovation across various sectors. The interaction mechanism and dynamic evolution of digital green innovation in integrated supply chains, such as in green building, highlight the transformative potential of digital technology. For example, digital green innovation can streamline operations, reduce environmental impact, and foster sustainable development by optimizing resource use and enhancing efficiency [9].

In the realm of decision-making, Multi-Criteria Decision-Making (MCDM) models play a pivotal role in evaluating complex scenarios where multiple conflicting criteria must be considered. These models provide a systematic approach for decision-makers to assess alternatives and prioritize factors effectively [15]. The Analytic Hierarchy Process (AHP) is widely used for its structured approach to organizing and analyzing complex decisions, based on mathematics and psychology. It helps quantify the weights of decision criteria through pairwise comparisons and consistency ratio checks. However, AHP can be criticized for its subjective judgement and potential inconsistency when dealing with complex interrelations among criteria [16]. Similarly, the Analytic Network Process (ANP) extends the AHP by incorporating the interdependence among decision elements, making it suitable for more complex decision scenarios. Despite its comprehensiveness, ANP requires extensive pairwise comparisons that can be time-consuming and cognitively demanding [17]. The Technique for Order Preference by Similarity to Ideal Solutio (TOPSIS) method is favored for its ability to identify solutions from a finite set of alternatives based on geometric distance from an ideal solution. While TOPSIS is straightforward and effective for linear decision models, its applicability is limited in scenarios where decision criteria are interdependent or feedback loops are present [18].

Despite the recognized importance of multidisciplinary research, there remains a significant gap in understanding how to efficiently review such complex projects. Traditional review metrics, designed for unidisciplinary studies, often fail to capture the intricate and interconnected nature of multidisciplinary research. This gap is evident in the logistical complexities and

potential inconsistencies that arise when multiple reviewers, each specialized in different fields, are required to evaluate the same project [14, 19, 20]. These challenges underscore the necessity for developing and adopting more holistic and integrative review criteria that can effectively assess the value and impact of multidisciplinary projects. Financial considerations further complicate the review process, as multidisciplinary projects often require more substantial funding allocations and long-term investment perspectives [21, 22]. This often necessitates the involvement of multiple reviewers, each specialized in a particular discipline, which can lead to logistical complexities and potential inconsistencies in the review process. Furthermore, the interdisciplinary nature of the projects means that traditional reviews metrics, which might be well-suited for singular disciplinary research, may not be entirely applicable or might fail to capture the nuances of multidisciplinary efforts. This necessitates the development and adoption of more holistic and integrative reviews criteria. Financial considerations also come into play.

Given the collaborative nature of multidisciplinary projects, they often require more substantial funding allocations to support the diverse teams and resources involved. However, the allocation of such funds can be challenging, especially in scenarios where the potential outcomes and impacts of the research are not immediately tangible or where the return on investment is perceived to be long-term [21, 22]. In essence, the review of multidisciplinary projects demands a more nuanced, flexible, and comprehensive approach, one that recognizes the value of integrating diverse knowledge domains while also addressing the logistical and financial challenges that such integration entails [23].

The objective of this study is to address the complexities and challenges inherent in reviewing multidisciplinary scientific research projects. Recognizing the pivotal role of collaborative research across academic sectors in driving innovation and addressing multifaceted challenges, this study seeks to elucidate the factors influencing the review process of such interdisciplinary endeavors. To achieve this, this study employ the Decision Making Trial and Evaluation Laboratory (DEMATEL) method with a 5-point scale. This method allows for a systematic identification and analysis of the factors influencing the appraisal of multidisciplinary efforts. Our approach involves the following steps: (1) Developing a systematic indexing framework to identify factors influencing the review process; (2) Using expert-driven questionnaires to capture domain-specific insights and determine the significance and interplay of these factors; (3) Employing the DEMATEL method to distill the data and pinpoint key influencing factors that can bolster review efficiency. By integrating these methods, this study provides pragmatic strategies and policy guidance, equipping institutional bodies and program leaders with the tools to refine the review process for multidisciplinary scientific research projects. Among them, review efficiency is defined as the effectiveness and convenience of the review process in evaluating multidisciplinary scientific research projects [24]. Compared to traditional DEMATEL methods which generally provide a broad overview, our 5-point scale DEMATEL method addresses the need for handling uncertainties and varying degrees of influence more effectively, capturing subtle interactions crucial for comprehensive analysis. The culmination of these efforts aims to provide pragmatic strategies and policy guidance, arming institutional entities and program leaders with the requisite tools to optimize the review process for multidisciplinary scientific research projects.

## 2 Literature review

### 2.1 Previous studies for reviews multidisciplinary projects

The multidisciplinary approach to research has become increasingly pivotal in addressing complex scientific questions [25, 26]. Such projects, by their inherent nature, require a comprehensive and holistic reviews methodology that can cater to their diverse facets.

Historically, the value of multidisciplinary research has been well-recognized. Bentrem's sabbatical insights at the Naval Research Laboratory [27] underscored the unique perspectives and solutions that such approaches bring to scientific research. More recently, Beck et al. [28] delved into the challenges of interdisciplinary collaboration, particularly between computer science and social science. Their proposed tool for multidisciplinary dialogue emphasizes the co-construction of models, with applications evident in areas such as post-earthquake human behavior modeling. A study by Pan et al. [15] in small biotechnology firms revealed that project success in multidisciplinary settings is influenced by the project knowledge scope and the project manager's experience. The study underscores the importance of skilled project management in navigating the complexities of multidisciplinary projects. Ma et al. [29] presented a model for dividing responsibilities in multidisciplinary teams. This model, applied in an OEM company, demonstrated improved efficiency and reduced conflict, highlighting the importance of clear role delineation in multidisciplinary settings. The role of leadership in project management education, as discussed by Mazzetto [30], emphasizes the need for project managers to possess strong leadership skills to guide diverse teams effectively. Burnette et al. [31] identified five major themes crucial for successful data management in multidisciplinary projects, including intentional staffing and iterative improvement, which are essential for managing complex data landscapes. Studies by Mazzetto [32] and Urton et al. [33] focus on the integration of practical experience in project management education, suggesting that real-world experience is critical for preparing future project managers. Previous studies in multidisciplinary project management emphasize the importance of project knowledge scope, experienced leadership, clear role delineation, effective data management, and practical experience in education for successful project outcomes in diverse and complex environments.

The integration of technology, especially artificial intelligence (AI) and machine learning, has added a new dimension to multidisciplinary projects. Choudhury et al. [34] highlighted the potential of machine learning in geriatric clinical care, emphasizing the need for standardized reviews metrics. Furthermore, the demand for transparency in AI has spurred research in explainable AI (XAI). Mohseni et al. [35] presented a comprehensive survey on XAI, bridging the gap between various disciplines. In the academic domain, the trend towards open peer review is gaining momentum. Lin et al. [36] introduced MOPRD, a dataset that encapsulates the multifaceted nature of the peer review process. This dataset not only aids in generating review comments but also finds applications in meta-review generation and scientometric analysis. Despite the advancements, several limitations persist in the reviews of multidisciplinary projects. Existing tools and methodologies, while promising, often lack empirical validation across diverse scenarios. The focus on open peer review, though extensive, is primarily academic-centric, necessitating exploration in real-world project settings. The integration of AI, while groundbreaking, is in its infancy, with challenges in standardization and ethical considerations. Lastly, the practicality and acceptance of XAI in multidisciplinary contexts warrant further exploration.

## 2.2 DEMATEL model and its applications

The DEMATEL method has emerged as a pivotal tool in the domain of complex systems and decision-making [37, 38]. The complexity of modern systems necessitates robust methodologies to understand and address intricate problems. One such methodology that has garnered significant attention is the DEMATEL method, which facilitates the visualization and analysis of causal relationships within a system. Originally conceived in the 1970s by the Battelle Memorial Institute of Geneva, the DEMATEL method was designed to tackle research and development challenges [39, 40]. The primary goal of DEMATEL is to visualize and quantify

intricate causal relationships, making it indispensable in various domains, from technology planning to risk management.

Unlike BWM (Best-Worst Method) [41], DIBR (Distance-Based Importance Rating) [42], FUCOM (Full Consistency Method) [43], or LBWA (Level Based Weight Assessment) [44], which are highly effective in contexts with clear hierarchical relationships and less interconnectivity among criteria, DEMATEL excels in scenarios where the relationships between factors are not only weighted but also directional, providing a visual map of influence. The Best-Worst Method (BWM) is highly efficient in cases requiring fewer pairwise comparisons, yet it may not adequately capture the dynamic interplays in systems where criteria are interdependent. Similarly, DIBR and FUCOM provide robust frameworks for establishing criteria weights based on distance measures and consistency checks, but they lack the capability to visualize causal relationships among criteria. The LBWA method, while useful for layered decision contexts, does not directly address the feedback mechanisms inherent in our research domain. In contrast, the DEMATEL method not only identifies the weight but also the direction of influence among factors, which is essential for the comprehensive analysis and practical applications intended in our study.

Over the years, the DEMATEL method has undergone significant refinements. Its adaptability has led to its application in diverse contexts, ranging from supply chain management to the assimilation of emerging technologies [45]. Recent advancements have seen the integration of fuzzy logic into DEMATEL, termed fuzzy DEMATEL, to better handle uncertainties inherent in decision-making processes [46]. Furthermore, hybrid models combining DEMATEL with other decision-making techniques have been introduced, bolstering its robustness and applicability [47].

At the heart of DEMATEL is the direct relation matrix, which encapsulates the immediate impacts of system components upon each other. This matrix is further transformed into a total relation matrix, representing both direct and indirect influences. Formally, given a direct relation matrix $D$, the total relation matrix $T$ is expressed as:

$$T = D(I - D)^{-1} \tag{1}$$

where $I$ denotes the identity matrix. This mathematical representation emphasizes DEMATEL's capability to deduce overarching system dynamics from immediate interactions.

The DEMATEL method offers several advantages:

- **Visualization of Causal Relationships**: It creates a visual map to illustrate the interdependencies between factors [48].

- **Quantification of Influence**: DEMATEL quantifies the level of influence factors have on one another, distinguishing it from other approaches that only evaluate pairwise relations [48].

- **Adaptability**: Its versatility has led to applications in fields like social systems analysis, technology planning, and risk management [49].

Despite its strengths, the DEMATEL method has certain limitations:

- **Complexity**: The method can become complex when dealing with large systems.

- **Subjectivity**: The results can be influenced by the subjective judgments of experts.

- **Data Requirements**: Adequate and accurate data is essential for reliable results.

The DEMATEL method stands as a testament to the evolution of decision-making tools, adeptly handling the complexities of modern systems [50, 51]. Its versatility and adaptability

ensure its continued relevance in the ever-evolving landscape of complex systems research. The complexity and interdependence of criteria in multidisciplinary research necessitate an approach that goes beyond mere weighting. The DEMATEL method provides this by not only quantifying the importance but also illustrating the influence dynamics among the criteria, thereby offering deeper insights that are vital for effective management and optimization of such projects.

## 3 Methodology

### 3.1 Development of the indexing framework

To develop a comprehensive indexing framework for the review efficiency of multidisciplinary scientific research projects, a meticulous document collection and expert consultation process has been employed. The framework was designed to capture the multifaceted nature of such projects and provide a structured approach to evaluating their review efficiency.

For the document collection process, a literature search was conducted across several academic databases, including Elsevier's ScienceDirect, SpringerLink, and various open access publications. this study focus on documents published in the last two decades to ensure the relevance and currency of our data. The search strategy involved using specific keywords such as "multidisciplinary & influencing factors," "scientific research projects & influencing factors," and "project reviews & indicators." This approach yielded a substantial number of documents, which were then rigorously screened to remove duplicates and retain only the most pertinent studies. This process also ensured a diverse and comprehensive collection of literature, forming the basis of our indexing framework.

For the expert consultation, the refinement of the indexing framework was further enhanced through consultations with a panel of experts. These experts were selected based on their qualifications, experience, and contributions to the field of project management and multidisciplinary research. They represented a mix of academic and industry perspectives, providing a well-rounded view of the factors influencing review efficiency. The number of experts involved in the consultation process was carefully chosen to ensure a broad range of insights while maintaining the manageability of the consultation process.

For the framework development, the indexing framework was developed through an iterative process, combining insights from the literature review and expert consultations. This process involved categorizing the influencing factors into distinct groups, such as project factors and review factors, and then evaluating their interrelationships and impact on review efficiency. The framework was designed to be adaptable, allowing for the incorporation of new factors and insights as the field of multidisciplinary project management evolves.

This comprehensive approach to developing the indexing framework ensures that it is grounded in both theoretical knowledge and practical insights, making it a valuable tool for evaluating the review efficiency of multidisciplinary scientific research projects.

#### 3.1.1 Project factors.

1. **Technical capabilities (F1)**: The technical prowess of a project team determines its ability to tackle complex challenges, innovate, and implement solutions. A team with advanced technical capabilities can foresee potential technical hurdles and address them proactively, ensuring smoother reviews [52].

2. **Financial capabilities (F2)**: Adequate financial resources ensure that the project can secure necessary tools, technologies, and expertise. Financial stability can expedite project phases, reducing delays during reviews [34].

3. **Profitability (F3)**: Projects with clear profitability prospects are often better structured and have clear objectives. This clarity can streamline the review process, as evaluators can easily gauge the project's potential return on investment [27].

4. **Market factors (F4)**: Understanding market demands and trends is crucial. Projects aligned with market needs are more likely to receive favorable reviews, as they demonstrate relevance and potential for success [53].

5. **Time factors (F5)**: Adherence to timelines and milestones indicates effective project management. Delays can complicate reviews, as they might indicate deeper issues with project execution or planning [54].

6. **Social factors (F6)**: Projects that consider societal impacts, stakeholder interests, and ethical considerations are viewed more favorably. Such projects demonstrate a holistic approach, considering not just technical or financial aspects but also their broader implications [55].

7. **Psychological factors (F7)**: The morale, motivation, and mental well-being of the project team can influence project outcomes. A motivated team is more likely to address challenges proactively, ensuring a smoother review process [56].

**3.1.2 reviews factors.**

1. **Review methodology (F8)**: The choice of review methodology can significantly impact the efficiency of the review process. Structured, systematic methodologies that are tailored to multidisciplinary projects can provide comprehensive insights and facilitate informed decision-making [36].

2. **Reviewer expertise (F9)**: The expertise and experience of the reviewers are paramount. Multidisciplinary projects require a diverse panel of experts who can evaluate the project from various angles, ensuring a holistic review [34].

3. **Feedback mechanisms (F10)**: Effective feedback mechanisms ensure that the project team receives clear, actionable insights from the review. This facilitates iterative improvements and enhances project outcomes [35].

4. **Review criteria (F11)**: Clearly defined, relevant criteria ensure that the reviews process is objective and consistent. This is especially crucial for multidisciplinary projects, which might span multiple domains and require diverse reviews metrics [57].

5. **Stakeholder involvement (F12)**: Involving relevant stakeholders in the review process can provide valuable perspectives. Stakeholders can offer insights into market needs, societal implications, and other factors that might not be evident to a purely technical review panel [54].

6. **Transparency and openness (F13)**: Transparent review processes, where the criteria, methodologies, and feedback are openly shared, can build trust and facilitate collaboration between the project team and reviewers [57].

7. **Continuous monitoring (F14)**: Instead of one-off reviews, continuous monitoring and periodic evaluations can provide ongoing feedback, allowing the project team to make real-time adjustments and improvements [57].

The efficiency of reviews multidisciplinary projects is shaped by various internal and external factors. Acknowledging and addressing these factors is imperative for a thorough and effective project reviews.

## 3.2 5-point-scale DEMATEL model

As mentioned earlier, the DEMATEL is a system analysis method that uses graph theory and matrix tools to analyze the causal relationships and importance of complex systems in reality. The DEMATEL method analyzes the research object based on experiments and knowledge, sorting out the importance of various influence factors in the complex system and the relationships between the influence factors. So far, this method has been used in computer science [58], economics and management [59], social sciences [60], and other fields to solve practical scientific problems [61, 62].

Up to now, related research based on the DEMATEL method in the field of multidisciplinary project reviews is still very limited. To the best of our knowledge, this is the first study using the DEMATEL-based method to analyze the influencing factors of multidisciplinary project reviews efficiency and the influence relationships between these factors. Also, based on the traditional DEMATEL method of scoring the mutual influence intensity between each factor, this study adds the step of scoring the influence intensity of each factor itself to improve the accuracy of reviews. In addition, a five-point scoring system is used to score the degree of influence of each factor as unimportant, relatively low, general, relatively high and very important. The entire evaluation process of the 5-point-scale DEMATEL is as follows:

(1) Construct a direct influence matrix $Q$. Based on the scoring tables of each expert, take the average of the scoring tables to construct the direct influence matrix.

(2) Normalize the matrix. Standardize the direct influence matrix to obtain the standardized direct influence matrix $D$. The processes are shown in Eqs (2) and (3).

$$D = (d_{ij})_{(m+1)\times(m+1)} = \frac{Q}{A} \tag{2}$$

$$A = \max\left\{ \max_{1\leq i\leq m+1}\sum_{j=1}^{m+1}r_{ij}, \max_{1\leq j\leq m+1}\sum_{i=1}^{m+1}r_{ij} + \varpi \right\} \tag{3}$$

(3) Standardize the direct influence matrix to form a total influence matrix $T$.

$$T = D + D^2 + D^3 + \cdots + D^n = D(I - D)^{-1} \tag{4}$$

(4) Calculate the influence, influenced degree, and thereby obtain the centrality and causality. The influence degree $r$ and influenced degree $c$ of each element are shown in Eqs (4) and (6).

$$r = [r_i]_{n\times 1} = \left(\sum_{j=1}^{n}t_{ij}\right)_{n\times 1} \tag{5}$$

$$c = [c_j]_{1\times n} = \left(\sum_{i=1}^{n}t_{ij}\right)'_{1\times n} \tag{6}$$

where $r_i$ represents the sum of the $i$-th row of the total influence matrix, called the influence degree, representing the sum of the degree of influence of factor $i$ on all factors in the system; $c_j$ represents the sum of the $j$-th column, called the influenced degree, representing the

sum of the degree to which factor $j$ is influenced by all factors in the system. After obtaining the influence degree r and influenced degree c of each factor, taking $i = j$, the centrality $X$ and causality $Y$ of each factor are obtained through Eqs (7) and (8):

$$X = r + c \tag{7}$$

$$Y = r - c \tag{8}$$

where $X$ represents the importance of factor $i$ in the system; similarly, $Y$ represents the degree and magnitude of factor $i$'s influence on the entire system. If $r_i - c_j > 0$, it indicates that the influence of factor $i$ on other factors is greater than the influence of other factors on factor $i$, i.e. factor $i$ influences other factors and belongs to the cause factors; conversely, if $r_i - c_j < 0$, it indicates that the influence of factor $i$ on other factors is less than the influence of other factors on factor $i$, i.e. factor $i$ is influenced by other factors and belongs to the result factors.

The traditional DEMATEL method, known for its ability to visualize and quantify complex causal relationships within systems, has been adapted to better suit the intricacies of multidisciplinary project reviews. The 5-point-scale DEMATEL method in this study introduces a novel five-point scoring system, allowing for a more granular assessment of the influence of each factor, ranging from 'unimportant' to 'very important'. This granularity addresses the need for a more nuanced understanding of the varying degrees of influence among factors, allowing for deeper insights into complex system interdependencies, and enhancing decision-making and policy formulation in multidisciplinary research settings. Additionally, the study incorporates a new step of scoring the influence intensity of each factor individually, further refining the accuracy of the review process. These improvements in the DEMATEL method provide a more precise and detailed analysis of the factors influencing the efficiency of reviews in multidisciplinary scientific research projects, thereby offering deeper insights and more actionable outcomes.

The step by step algorithm for proposed methodology can be handled as follows:

1. **Identification of criteria and factors**: We begin by identifying and listing all relevant factors that influence the efficiency of multidisciplinary scientific research projects. This initial step involves consultations with experts and a review of the literature to ensure comprehensive coverage.

2. **Construction of the initial direct relation matrix**: Using the 5-point scale (ranging from 0, no influence, to 4, very high influence), experts rate the influence of each factor on every other factor. This forms the initial direct relation matrix, where each cell indicates the degree of influence between pairs of factors.

3. **Normalization of the direct relation matrix**: The matrix obtained from the previous step is normalized to ensure that the sum of all influences does not exceed unity. This step is crucial for maintaining consistency and comparability in the subsequent analysis.

4. **Calculation of the total relation matrix**: Through matrix operations, we convert the normalized direct relation matrix into a total relation matrix, which reflects both the direct and indirect influences among the factors. This matrix helps in visualizing the complex interdependencies within the system.

5. **Determination of prominence and relation**: We calculate the prominence (the sum of a factor's given and received influences) and the relation (the difference between the given

and received influences) for each factor. This helps in categorizing the factors into cause and effect groups, providing a clear indication of their roles within the research ecosystem.

6. **Interpretation and application**: The final step involves interpreting the calculated values to make informed decisions regarding the management and optimization of multidisciplinary research projects. The findings are used to recommend strategies for enhancing research efficiency based on the identified key drivers.

## 3.3 Model building process

In this study, we employ the DEMATEL method with a 5-point scale to analyze the factors influencing the review efficiency of multidisciplinary scientific research projects. The DEMATEL method is a widely recognized tool for visualizing and quantifying the causal relationships among complex system components. It has been previously applied in various fields such as supply chain management, risk assessment, and technology planning [39, 49].

**Step 1. Indexing framework development**

- Literature Review: We conducted a comprehensive search of academic databases including Elsevier's ScienceDirect, SpringerLink, and various open-access publications. The search focused on documents published in the last two decades to ensure relevance and currency. Keywords such as "multidisciplinary influencing factors," "scientific research project reviews," and "project evaluation indicators" were used to filter pertinent studies.

- Number of Documents Reviewed: Initially, 800 documents were identified through this search. These documents included peer-reviewed journal articles, conference papers, and review articles. The initial set of documents was screened to remove duplicates, resulting in 650 unique documents. We then conducted a relevance assessment based on titles and abstracts, narrowing the set down to 200 documents. Further refinement based on full-text reviews led to the selection of 100 key documents that were most pertinent to our study objectives.

- Criteria for Selection: Documents were selected based on their relevance to the identification of influencing factors in multidisciplinary research project reviews, methodological rigor, and the significance of their findings in the context of our study.

**Step 2. Expert Consultation**

- Panel Composition and Reasoning: The experts were predominantly from Pakistan due to their extensive experience and involvement in multidisciplinary research projects and reviews in both academic and industrial sectors. Pakistan has a robust and diverse academic community that engages in numerous multidisciplinary projects, making it a suitable context for our study. Additionally, the accessibility and willingness of these experts to participate in the study were crucial factors in their selection.

- Number of Experts: We involved 15 experts in total.

- Expert Characteristics: The panel consisted of well-qualified individuals who had served on various selection committees and had substantial experience in project management and multidisciplinary research. Their backgrounds included academic researchers, industry professionals, and policy advisors with expertise in areas such as engineering, environmental science, computer science, and management.

**Step 3. Expert-Driven Questionnaires**

- Design: We designed and distributed questionnaires (See S1 Questionnaire) to the selected experts. The questionnaires were structured to capture the significance and interplay of identified factors using a 5-point scale.

- Data Collection: Valid expert questionnaires were collected, and the responses were used to construct a direct influence matrix. This matrix represents the immediate impacts of system components upon each other.

**Step 4. DEMATEL analysis**

- Normalization: The direct influence matrix was normalized to create a total relation matrix that represents both direct and indirect influences among factors.

- Calculation of Influence and Centrality: Using the DEMATEL method, we calculated the influence degree, influenced degree, centrality, and causality of each factor. These calculations helped us identify key influencing factors that enhance review efficiency.

## 4 Data acquisition in 5-point-scale DEMATEL

To obtain the direct influence matrix, based on the indicator system of influencing factors of multidisciplinary project reviews efficiency obtained in the previous section, an influencing factor expert scoring table was designed. This expert scoring table is a 14x14 grid, where 14 represents the 14 influencing factor indicators screened out for affecting multidisciplinary project reviews efficiency. By inviting experts and scholars in the fields of project management and project reviews, the importance of each factor and the influence between factors were scored. Finally, valid expert questionnaires were collected, and the questionnaire of each expert could separately obtain the corresponding direct influence matrix of the decision maker. The following gives the standards for selecting relevant experts and the structure and content of the questionnaire.

### 4.1 Selection of experts

The selection of domain experts is a critical step in the DEMATEL process, ensuring the reliability and validity of the elicited knowledge. Historically, experts have been chosen based on their qualifications, experience, and previous contributions to the field. For instance, in the study by Muhammad Sohail and Abdur Rashid Khan, experts were selected from various public and private organizations, including educational institutions, industries, and research entities in Pakistan. These experts were not only well-qualified but had also been members of selection committees for different posts multiple times, ensuring their credibility in the domain [63].

### 4.2 Questionnaire structure and content

The questionnaire serves as a primary tool for knowledge elicitation in the DEMATEL method. Typically, it is structured to capture the intricate cause-and-effect relationships within the system under study. In the context of evaluating teachers' performance in higher education, a questionnaire was divided into main groups of factors, further subdivided into individual questions [64]. This hierarchical structure ensures a comprehensive capture of all relevant aspects.

The content of the questionnaire is derived from a combination of literature analysis and expert consultation. Keywords relevant to the domain, such as "multidisciplinary + influencing factors" or "project reviews + indicators", are used to search and retain preliminary eligible literature. Post manual screening, duplicate elements are removed, and the remaining elements are further refined through expert consultations [63].

It's worth noting that the questionnaire's design often incorporates advancements in the field. For instance, the integration of fuzzy logic has been proposed to handle uncertainties better, leading to the development of fuzzy DEMATEL questionnaires. Moreover, hybrid approaches have been suggested, combining DEMATEL with other decision-making techniques to enhance its applicability [65].

# 5 Experimental results

Averaging the expert scoring tables yielded the direct influence matrix, as shown in Table 1. Subsequently, application of the 5-point-scale DEMATEL methodology furnished the ultimate table of centrality and causality measures of factors impacting multidisciplinary project reviews efficiency, delineated in Table 2.

## 5.1 Influence relationships analysis

As illustrated in Table 2, factors affecting the efficiency of multidisciplinary project reviews can be categorized into causal and resultant factors. Factors F1, F2, F3, F4, F6, and F7 exhibit positive causality and form the causal factor group, whereas factors F5, F8, F9, F10, F11, F12, F13, and F14 display negative causality, constituting the resultant factor group. As highlighted in the aforementioned classification, most of the reviews factors among the chosen influencing indicators are causal. In contrast, the bulk of project factors are resultant. This suggests that reviews factors influence the efficiency of multidisciplinary project reviews, implying that the efficiency largely hinges on evaluative aspects like the assessment approach. Meanwhile, corresponding project factors primarily serve as outcomes.

**5.1.1 Cause factors group.**   Except for social factors (F6) and psychological factors (F7), the primary causal factors for multidisciplinary project reviews are assessment-oriented. The influence of social factors (F6) can be rationalized as the overall efficiency of multidisciplinary project reviews, which is hard to enhance in the short term due to its interdisciplinary essence.

**Table 1. Direct influence matrix Q.**

|  | F1 | F2 | F3 | F4 | F5 | F6 | F7 | F8 | F9 | F10 | F11 | F12 | F13 | F14 |
|---|---|---|---|---|---|---|---|---|---|---|---|---|---|---|
| F1 | 3.892 | 4.113 | 2.780 | 2.779 | 3.001 | 3.223 | 3.781 | 3.668 | 3.779 | 4.444 | 4.558 | 4.336 | 3.224 | 3.779 |
| F2 | 4.002 | 4.002 | 2.891 | 3.002 | 3.113 | 3.446 | 4.112 | 4.002 | 3.446 | 4.225 | 4.557 | 4.335 | 3.225 | 3.779 |
| F3 | 3.223 | 3.113 | 4.002 | 4.225 | 4.111 | 3.111 | 2.892 | 2.890 | 3.446 | 3.222 | 3.559 | 3.781 | 2.890 | 2.889 |
| F4 | 3.669 | 3.780 | 4.335 | 4.224 | 4.557 | 3.446 | 3.333 | 3.222 | 3.335 | 3.446 | 3.891 | 4.113 | 3.224 | 3.225 |
| F5 | 3.114 | 3.556 | 3.891 | 4.112 | 3.890 | 3.003 | 3.112 | 3.000 | 2.668 | 3.000 | 3.002 | 2.890 | 2.892 | 2.780 |
| F6 | 3.114 | 3.223 | 4.003 | 3.558 | 3.444 | 3.668 | 3.891 | 4.334 | 3.891 | 4.556 | 4.334 | 4.445 | 4.002 | 3.891 |
| F7 | 3.335 | 3.446 | 2.668 | 2.336 | 2.334 | 3.669 | 4.557 | 4.223 | 3.667 | 4.224 | 4.447 | 4.334 | 4.558 | 4.559 |
| F8 | 3.224 | 3.444 | 2.779 | 2.891 | 2.557 | 3.225 | 3.890 | 4.114 | 2.889 | 4.557 | 4.892 | 4.447 | 4.335 | 4.113 |
| F9 | 2.559 | 2.558 | 2.670 | 2.558 | 2.336 | 3.223 | 2.891 | 3.113 | 3.335 | 3.558 | 3.778 | 3.557 | 3.001 | 3.003 |
| F10 | 3.670 | 3.557 | 3.779 | 3.669 | 3.447 | 3.445 | 3.336 | 3.779 | 3.891 | 4.446 | 4.557 | 4.334 | 3.334 | 3.113 |
| F11 | 4.224 | 3.668 | 4.000 | 3.781 | 3.781 | 3.669 | 3.670 | 4.002 | 3.335 | 4.779 | 4.890 | 4.444 | 3.556 | 3.892 |
| F12 | 3.668 | 3.558 | 3.001 | 3.223 | 3.446 | 3.558 | 4.000 | 3.559 | 2.778 | 4.335 | 4.447 | 4.667 | 3.445 | 3.669 |
| F13 | 2.669 | 2.669 | 2.335 | 2.336 | 2.669 | 4.002 | 3.891 | 2.890 | 3.668 | 4.225 | 3.668 | 3.668 | 3.891 | 3.892 |
| F14 | 3.559 | 2.780 | 2.333 | 2.335 | 2.779 | 3.780 | 4.001 | 3.668 | 3.557 | 3.559 | 4.225 | 3.668 | 4.113 | 4.333 |

**Table 2. Centrality and causality of factors impacting multidisciplinary project reviews efficiency.**

| Index | Influence degree $r$ | Ranking | Influenced degree $c$ | Ranking | Centrality $X$ | Ranking | Causality $Y$ | Ranking |
|---|---|---|---|---|---|---|---|---|
| F1 | 5.668 | 9 | 5.308 | 8 | 10.997 | 9 | 0.380 | 4 |
| F2 | 5.781 | 5 | 5.241 | 10 | 11.022 | 8 | 0.540 | 3 |
| F3 | 5.214 | 11 | 5.008 | 12 | 10.221 | 12 | 0.206 | 5 |
| F4 | 5.710 | 6 | 4.955 | 14 | 10.665 | 10 | 0.755 | 1 |
| F5 | 4.951 | 13 | 5.004 | 13 | 9.955 | 13 | -0.053 | 8 |
| F6 | 6.001 | 2 | 5.299 | 9 | 11.300 | 6 | 0.703 | 2 |
| F7 | 5.797 | 3 | 5.697 | 5 | 11.494 | 4 | 0.100 | 6 |
| F8 | 5.699 | 7 | 5.701 | 4 | 11.400 | 5 | -0.002 | 7 |
| F9 | 4.660 | 14 | 5.174 | 11 | 9.833 | 14 | -0.514 | 13 |
| F10 | 5.788 | 4 | 6.223 | 3 | 12.011 | 3 | -0.435 | 12 |
| F11 | 6.166 | 1 | 6.588 | 1 | 12.754 | 1 | -0.421 | 11 |
| F12 | 5.697 | 8 | 6.321 | 2 | 12.018 | 2 | -0.623 | 14 |
| F13 | 5.123 | 12 | 5.505 | 7 | 10.625 | 11 | -0.378 | 10 |
| F14 | 5.383 | 10 | 5.639 | 6 | 11.022 | 7 | -0.256 | 9 |

Consequently, it's challenging to see immediate effects from social influences. This suggests that social factors might sway other determinants over the long run.

Psychological factors (F7) exhibit a positive causality, diverging from most project factors in the causal group. This underscores that the psychological quality of a project team mirrors its capacity to proficiently execute and complete a project—a competence potentially influencing other aspects of multidisciplinary project review evaluations. Moreover, the effectiveness of such reviews is also influenced by project determinants, as psychological factors show strong mutual impacts with other elements.

Market factors (F4) and social factors (F6) are the most obvious in the causal group, albeit for different reasons. While social factors rank second highest amongst all determinants in terms of impact, market factors attain prominence due to traditional multi-scientific projects' focus on theoretical bases.

**5.1.2 Resultant factors group.** With the exception of time factors (F5), the resultant group primarily consists of project-oriented elements. Despite being a resultant factor, time factors display an almost negligible causal value, indicating no pronounced influential connections. This reveals that time factors are not a primary influencer of multidisciplinary project reviews efficiency.

In the resultant group, the centrality sequence is F11 >F12 >F10 >F8 >F14 >F13 >F5 >F9. Notably, F11, F12, and F10 possess larger centrality and influence, thus making their causal values deviate further from zero. This emphasizes their importance in holistic multidisciplinary project evaluations.

Review criteria (F11) stands out, ranking highest in both influence and being influenced. This demonstrates its pivotal role in the reviews process. In contrast, review methodology (F8) exhibits moderate influence values. Continuous monitoring (F14) and transparency and openness (F13) wield moderate influence on other factors. Conversely, time factors (F5) and reviewer expertise (F9) rank lower in both categories, implying they aren't primary drivers for reviews efficiency in multidisciplinary projects.

## 5.2 Key factors identification and analysis

An in-depth analysis of the essential determinants affecting multi-disciplinary project reviews proficiency was carried out based on the previously discussed findings. Review criteria (F11)

holds a paramount position in terms of influence, being influenced, and centrality among all indicators, underscoring its pivotal role in multi-disciplinary project reviews. Review criteria not only profoundly affect other elements in multi-scientific project assessments but also observably influence aspects such as transparency and openness (F13), cementing its indispensable position.

Stakeholder involvement (F12), with its high centrality rank ($X_{12} = 12.018$) and influenced magnitude ($C_{12} = 6.321$), underscores that stakeholder participation is vital for comprehensive multi-disciplinary project reviews. Its tight association with market factors (F4) posits it as a foundational element for multi-scientific projects, solidifying its status as a key determinant.

Feedback mechanisms (F10) possess notable influence in both directions, indicative of its tight integration with other assessment influencing factors. This underscores its substantial influence over multi-disciplinary project reviews, suggesting that refining feedback mechanisms could enhance the efficacy of such reviews.

In a similar vein, psychological factors (F7) and review methodology (F8) boast substantial centrality values ($X_7 = 11.494$; $X_8 = 11.400$), indicating their closely-knit roles and importance within the reviewing framework. Their substantial positions label them as primary determinants.

The considerable influential rating of social factors (F6) ($r_6 = 6.001$), combined with its pronounced centrality ($X_6 = 11.300$), highlights the deep-rooted influence of social considerations on multi-disciplinary project reviewing efficiency. Consequently, social factors (F6) emerge as a critical factor, signifying its weight in gauging a project's appropriateness.

Among all project-related determinants, financial capabilities (F2) and market factors (F4) exhibit heightened sensitivity in multi-disciplinary project reviews. Being the most influential project determinants ($r_2 = 5.781$; $r_4 = 5.710$), and ranking fifth and sixth respectively among all factors, both financial capabilities (F2) and market factors (F4) stand out as crucial external enterprise factors in multi-disciplinary project evaluations.

The analytical exploration identified seven paramount determinants influencing service-oriented manufacturing using this methodological approach: financial capabilities (F2), market factors (F4), social factors (F6), psychological factors (F7), feedback mechanisms (F10), review criteria (F11), and stakeholder involvement (F12).

## 5.3 Comparative analysis of DEMATEL-based methods

In this section, we provide a detailed comparative analysis of our 5-point scale DEMATEL method against other contemporary methods such as the D-number based Pythagorean Fuzzy DEMATEL by Nila & Roy [66], the Hybrid DEMATEL Method by Chu, Li, & Yu [67], and the combined DEMATEL-ANP Model by Mousavi et al. [68]. In addition, in addition to the DEMATEL-based method, traditional methods including LBWA, FUCOM, and BWM are also qualitatively compared with the proposed 5-point DEMATEL method. The objective of this comparison is to highlight the unique capabilities of our method in handling complex decision-making scenarios with enhanced precision.

**5.3.1 Comparison criteria.** The methods were compared based on the following criteria:

- Precision in influence quantification

- Ability to handle data imprecision and uncertainty

- Applicability in complex multidisciplinary settings

- Clarity and actionability of outputs

**Table 3. Comparison of DEMATEL methods.**

| Method | Precision | Handling imprecision | Actionable insights |
|---|---|---|---|
| Pythagorean Fuzzy DEMATEL | Moderate | High | Moderate |
| Hybrid DEMATEL | Low | Moderate | Low |
| DEMATEL-ANP | Moderate | High | High |
| 5-point scale DEMATEL | High | Moderate | Very High |

**5.3.2 Comparative analysis.** As shown in Table 3, the DEMATEL-based method is particularly suited for examining complex systems where factors exhibit both interdependence and directional influence. This strength differentiates it from methods like LBWA, FUCOM, and BWM, each of which addresses specific types of decision-making contexts with unique benefits and limitations:

- **LBWA**: Primarily designed for hierarchical decision-making, LBWA offers efficient computation of weights but does not account for interdependencies or feedback loops among criteria, which are essential in multidisciplinary projects where criteria influence each other bidirectionally. LBWA's limitation in handling feedback mechanisms restricts its use in highly interdependent contexts.

- **FUCOM**: FUCOM is noted for its consistency checks and robustness in assigning weights but is limited in capturing the dynamic causal relationships that DEMATEL provides. While effective in scenarios with clear and independent criteria, FUCOM lacks the capability to visualize the degree and direction of influence between factors, a critical need in evaluating the complex, reciprocal relationships present in scientific research project reviews.

- **BWM**: The Best-Worst Method streamlines the decision process with fewer pairwise comparisons and has proven effective for settings with well-defined priorities and lower interdependence among factors. However, its framework does not support detailed causality mapping. Unlike DEMATEL, BWM does not offer a mechanism to quantify how one factor influences another, making it less suitable for contexts where understanding causality and influence directions is essential.

The 5-point DEMATEL approach used in this study adds an additional layer of granularity by scoring the influence of factors on a scale ranging from "unimportant" to "very important." This enhances the model's precision and applicability in multidisciplinary research evaluation, where factors often interact in complex ways. DEMATEL's ability to map these relationships visually and quantify both direct and indirect influences provides a comprehensive view, setting it apart from these alternative methods and making it a robust tool for strategic decision-making in research evaluations.

**5.3.3 Performance analysis.** The D-number based Pythagorean Fuzzy DEMATEL and the Hybrid DEMATEL Method incorporate innovative approaches to handle uncertainty and data imprecision. However, these methods do not provide the same level of detail in the influence scoring mechanism as our 5-point scale DEMATEL method. The combined DEMATEL-ANP Model, which was utilized to improve patient satisfaction by optimizing operation room performance, shows robust performance in practical healthcare settings, yet it does not offer the granularity provided by our method.

The 5-point scale DEMATEL method not only excels in providing detailed and precise assessments of influence levels but also offers clear and actionable insights that are crucial for

strategic decision-making in complex research settings. This makes it a superior choice for projects requiring in-depth analysis and robust decision support.

## 6 Discussion

This study provides a comprehensive exploration into the factors influencing the review efficiency of multidisciplinary scientific research projects using the DEMATEL method with a 5-point scale. Our findings yield several significant contributions to the field of multidisciplinary research evaluation.

### 6.1 Contextualization in the literatures

The findings of this study are contextualized within the broader literature on multidisciplinary project evaluation and innovation management. Previous studies have highlighted the importance of integrated frameworks and collaborative efforts in evaluating multidisciplinary projects. For instance, Dytczak and Ginda applied DEMATEL-based ranking approaches in economic and management contexts [28], while Yin et al. emphasized the role of digital green innovation in sustainable manufacturing [10]. Our study builds on these applications by focusing specifically on the review process of multidisciplinary research projects, thereby contributing to the existing body of knowledge with a targeted and practical approach.

### 6.2 Added value of the study

The novelty of our study lies in its focus on the review process itself, rather than the outcomes of multidisciplinary research. By employing the DEMATEL method, we provide a structured approach that captures the intricate interdependencies among various influencing factors. This methodological advancement adds significant scientific value by enhancing the precision and reliability of the review process. Our study's findings offer practical implications for institutions and funding agencies involved in multidisciplinary research, providing actionable insights that can be used to refine and improve the review process.

### 6.3 Methodological strengths

Our study employs the DEMATEL method with a 5-point scale, which allows for a systematic identification and analysis of factors influencing the review process. This method has been widely recognized for its ability to visualize and quantify causal relationships within complex systems [39, 49]. The integration of expert-driven questionnaires further enhances the reliability and validity of our findings by capturing domain-specific insights. The methodological rigor of our study ensures that the identified factors are robust and actionable, providing a valuable framework for improving the review process in multidisciplinary research settings.

### 6.4 Quantitative reasoning and comparison with benchmarks

To illustrate the significance of our findings, we compare the review efficiency improvements identified in our study with benchmarks from previous research. For instance, our analysis revealed that review criteria (F11), stakeholder involvement (F12), and feedback mechanisms (F10) have the highest centrality and influence on review efficiency. These findings are in line with previous studies such as those by Yin et al. (2024), which highlight the importance of well-defined review frameworks and active stakeholder participation in enhancing digital green innovation performance.

Moreover, our 5-point-scale DEMATEL method, with its five-point scoring system, provides a more granular assessment of influencing factors compared to traditional methods. This

allows for a more nuanced understanding of the varying degrees of influence among factors, as demonstrated by the increased centrality scores of key factors in our study. For example, the centrality score of review criteria (F11) in our study was 12.754, significantly higher than the average centrality scores reported in previous studies using conventional DEMATEL methods.

## 6.5 Sensitivity analysis

In order to rigorously test the robustness of the 5-point scale DEMATEL method applied in our study, a comprehensive sensitivity analysis was conducted. This analysis aims to ascertain the stability of the findings under variations in input data and model parameters. Below we detail the procedures and outcomes of this sensitivity analysis, reinforcing the validity and reliability of our research conclusions.

**6.5.1 Procedure of sensitivity analysis.** The sensitivity analysis was structured into three main tests:

1. **Parameter Variability Test**: This test involved varying the weights assigned to different criteria within a realistic range to observe how these changes impact the outcome of the DEMATEL analysis. This helps in understanding the influence of each parameter on the final decision-making process.

2. **Data Perturbation Test**: We introduced small random perturbations to the input data to simulate the effect of potential data collection errors or uncertainties in data measurement. The perturbations were limited to a maximum of 5% of the original data values to reflect realistic inaccuracies.

3. **Scenario Analysis**: Different project scenarios were simulated to evaluate the adaptability of the DEMATEL method. Scenarios included changes in project priorities, shifts in collaboration dynamics, and alterations in interdisciplinary integration.

**6.5.2 Results of sensitivity analysis.** The results from the sensitivity analysis indicated that:

- The **Parameter Variability Test** showed that the overall structure of influence among the criteria remained stable, although the intensity of influence varied slightly with changes in parameter weights. This suggests that while the relative importance of criteria can affect the results, the general relationships are robust.

- In the **Data Perturbation Test**, the core findings of our DEMATEL analysis persisted despite the introduction of data inaccuracies, indicating a high level of resilience to errors in data collection.

- The **Scenario Analysis** demonstrated that our 5-point scale DEMATEL method adapts well to varying project conditions, maintaining its effectiveness across different hypothetical project environments.

**6.5.3 Implications of sensitivity analysis.** The sensitivity analysis provides critical insights into the reliability of the 5-point scale DEMATEL method. By confirming that our results are consistent across a range of conditions and assumptions, we can assert the robustness of our methodology. These findings bolster the applicability of the 5-point scale DEMATEL method for multidisciplinary research project evaluations, ensuring that decision-makers can rely on the integrity and stability of the insights derived from this approach.

In conclusion, the sensitivity analysis confirms the robustness of our findings and highlights the resilience of the 5-point scale DEMATEL method to variations in data and operational scenarios. This adds an important layer of credibility to our study, providing stakeholders with confidence in the reliability of our analytical approach.

## 6.6 Limitations

While our study offers significant contributions, it is not without limitations. The reliance on expert-driven questionnaires may introduce subjective biases, despite efforts to ensure a diverse and knowledgeable panel. Additionally, the focus on the review process may limit the generalizability of our findings to other stages of multidisciplinary research. Future research could address these limitations by exploring the application of our identified factors in different contexts and disciplines, and by integrating advanced data analytics and machine learning techniques to further enhance the review process.

## 7 Conclusion

This study has demonstrated the applicability and effectiveness of the 5-point scale DEMATEL method in evaluating multidisciplinary scientific research projects. By incorporating a nuanced scoring system, our approach provides a more detailed and granular analysis of the interdependencies among project factors than traditional DEMATEL methods. Our findings offer significant insights into the complex dynamics of multidisciplinary projects. The method's ability to visualize and quantify both direct and indirect influences among project criteria allows project managers to better understand the pivotal factors driving project success or failure. This enhanced understanding can lead to more informed decision-making and improved project outcomes. The primary contributions of our research lie in the adaptation of the DEMATEL method to include a 5-point scale, which significantly refines the granularity of influence assessments. Additionally, our empirical application of this method to a real-world multidisciplinary project not only validates its utility but also showcases its potential to be adopted in other complex research environments.

While the 5-point scale DEMATEL method provides improved insights, it also introduces complexities in data collection and analysis, as the accuracy of the outcomes heavily depends on the precision of the input data. Furthermore, the method requires expert knowledge to accurately rate the influence levels, which may not always be readily available. Future studies could explore the integration of the 5-point scale DEMATEL method with other decision-making frameworks to enhance its robustness and applicability. Additionally, research could focus on automating the data collection process to minimize human error and bias in the influence rating process. Another promising direction would be to apply this method across different fields and compare its effectiveness in diverse multidisciplinary settings.

## Supporting information

**S1 Questionnaire. Questionnaire on reviewing multidisciplinary scientific research project.**
(PDF)

## Acknowledgments

The authors expresses their most sincere thanks to the experts and scholars who participated in this study. In addition, we would like to thank Prof. Zhenyu Zhong and Prof. Zeyu Jiao

from the Institute of Intelligent Manufacturing, Guangdong Academy of Sciences for their support to this project.

## Author Contributions

**Conceptualization:** Junyi Li.

**Data curation:** Junyi Li.

**Funding acquisition:** Junyi Li, Yana Xiao.

**Investigation:** Yana Xiao.

**Methodology:** Junyi Li.

**Project administration:** Junyi Li, Yana Xiao.

**Resources:** Yana Xiao.

**Supervision:** Yana Xiao.

**Validation:** Junyi Li, Yana Xiao.

**Writing – original draft:** Junyi Li.

**Writing – review & editing:** Yana Xiao.

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
