## [Decision Letter · Decision Letter 0]

14 Dec 2023

PONE-D-23-29808Analysis of influencing factors of review efficiency of multidisciplinary scientific research projects based on improved DEMATEL modelPLOS ONE

Dear Dr. Xiao,

Thank you for submitting your manuscript to PLOS ONE. After careful consideration, we feel that it has merit but does not fully meet PLOS ONE’s publication criteria as it currently stands. Therefore, we invite you to submit a revised version of the manuscript that addresses the points raised during the review process.

 Below you may find enclosed Reviewers' comments and suggestion. First Reviewer points out to the weak theoretical basis and research framework.  While novelty of the study is not required by our Journal, sound research framework is the critical issue. Both Reviewers agree that this area need to be improved in your manuscript. Second Reviewer brings more details to your attention such as the lack of information on the collection of documents that was used for the search of indicators or on the experts that were consulted on the influencing factors. They also suggest to point out some factor that may be considered as potential limitations of your study, such as focus on applied research projects. These are only most important issues raised, you may find the rest below, in the Review reports.Having read the manuscript myself I have to suggest you some additional changes that will improve it's clarity. I will list them now:- I believe there is a word missing at the beginning of line 35- in my opinion part "2.1 Previous studies for reviews multidisciplinary projects" is too focused on AI. It needs either more adequate title or slight change of focus so that AI will not dominate the discussion here- reference in line 111 is missing- speaking of adaptability of  the DEMATEL method, you should include some references or examples (lines 143-145)- how do you define "reviews efficiency"? It is crucial for understanding the study- were "project factors" and "reviews factors" suggested by experts or Authors?- why mean was used in the study? Have you considered using median or mode?- "So far, this method has been used in computer science, economics and management, social sciences and other fields to solve practical scientific problems." - lines 218-219 - references or examples are needed here- you refer to "improved DEMATEL", but do not explain how it is improved as compared with the original version- in my opinion questionnaire should be attached as a supporting file- I do not understand the notation in line 329- I would refrain from using word "significant" since it suggest statistical test were run.

We look forward to receiving your revised manuscript.

Kind regards,

Agata Sielska, Ph.D.

Academic Editor

PLOS ONE

Journal Requirements:

2. You indicated that ethical approval was not necessary for your study. We understand that the framework for ethical oversight requirements for studies of this type may differ depending on the setting and we would appreciate some further clarification regarding your research. Could you please provide further details on why your study is exempt from the need for approval and confirmation from your institutional review board or research ethics committee (e.g., in the form of a letter or email correspondence) that ethics review was not necessary for this study? Please include a copy of the correspondence as an ""Other"" file

"This research was funded by Guangdong Provincial Science and Technology Plan Project grant number 2020B1010010005."

"The authors were funded by Guangdong Provincial Science and Technology Plan Project grant number 2020B1010010005."

"The authors were funded by Guangdong Provincial Science and Technology Plan Project grant number 2020B1010010005."     

6. PLOS requires an ORCID iD for the corresponding author in Editorial Manager on papers submitted after December 6th, 2016. Please ensure that you have an ORCID iD and that it is validated in Editorial Manager. To do this, go to ‘Update my Information’ (in the upper left-hand corner of the main menu), and click on the Fetch/Validate link next to the ORCID field. This will take you to the ORCID site and allow you to create a new iD or authenticate a pre-existing iD in Editorial Manager. Please see the following video for instructions on linking an ORCID iD to your Editorial Manager account: https://www.youtube.com/watch?v=_xcclfuvtxQ.

Reviewers' comments:

Reviewer's Responses to Questions

**Comments to the Author**

1. Is the manuscript technically sound, and do the data support the conclusions?

Reviewer #1: Partly

Reviewer #2: Partly

2. Has the statistical analysis been performed appropriately and rigorously? 

Reviewer #1: No

Reviewer #2: N/A

3. Have the authors made all data underlying the findings in their manuscript fully available?

Reviewer #1: No

Reviewer #2: No

4. Is the manuscript presented in an intelligible fashion and written in standard English?

Reviewer #1: No

Reviewer #2: Yes

5. Review Comments to the Author

Reviewer #1: The theoretical basis for the mechanisms presented in this study is very weak and unconvincing. The novelty of this manuscript is very inadequate. There are many similar studies. In addition, the research purpose and research framework of the manuscript are unclear. There are unaddressed issues in the paper that bear on the central conclusions of the paper.

Reviewer #2: The main objective of the study is to find out/elucidate the review process of interdisciplinary projects and enhance efficiency of this process (lines 58-59, 61). The authors address this well motivated challenge using DEcision MAking Trial and Evaluation Laboratory (DEMATEL) method. Although the method was developed yet in the 1970s, it is becoming more and more popular in the recent years as it allows to identify prominence and roles played by particular factors in the cause-and-effect relationships in the complex systems of various types. The number of documents indexed in the Scopus database that contain the term “DEMATEL” in the title, in the abstract or among the keywords expanded in the last two decades. Until 2005 such documents were practically nonexistent, in 2011-2013 a number of such documents achieved a level of 100 per annum and this year the document count exceeded 800.

The authors follow a two-stage procedure to tackle the challenge. First, they search literature using the keywords “multidisciplinary and influencing factors”, “scientific research projects and influencing factors”, and “project reviews and indicators”. Then, in the course of consultations with experts in scientific research project management the final set of 14 factors was refined. They were assigned either to the category of the “project factors”:

F1) Technical capabilities,

F2) Financial capabilities,

F3) Profitability,

F4) Market factors,

F5) Time factors,

F6) Social factors,

F7) Psychological factors,

or to the category of “reviews factors”:

F8) Review methodology,

F9) Reviewer expertise,

F10) Feedback mechanisms,

F11) Review criteria,

F12) Stakeholder involvement,

F13) Transparency and openness,

F14) Continuous monitoring.

The manuscript doesn’t give any information on the collection of documents that was used for the search of indicators (Elsevier’s ScienceDirect, SpringerLink, open access publications of other science publishers?), in which years these documents were published, how many documents were browsed. The reader doesn’t know either who are the experts that were consulted on the influencing factors or how many experts were involved.

The inclusion of factors such as F3, F4, F6 and F12 suggests that the authors are interested in the applied research rather than in the theoretical one, although earlier they mention such outcomes of the research as “the emergence of innovative theoretical constructs” (lines 20-21) and pay attention to the possibility that “potential outcomes and impacts of the research are not immediately tangible” (lines 49-50). The legibility of the paper would increase if the authors clarified whether they are interested in applied research only (without referring to the theoretical research or mentioning this kind of research with a due disclaimer) or in both applied and theoretical research (in such case the applicability of factors F3, F4, F6 and F12 should be explained).

The factors F5 and F7, and to the lesser extent F10 and F14, indicate that the focus of the study is the interim review of the research projects rather than the projects that apply for funding. It would be useful to clarify this issue.

The experts provided also the scoring tables that were used to calculate the direct influence matrix and the standardized matrix. Then the total influence matrix was computed and the indicators of centrality/prominence and causality were evaluated. According to these indicators the reviewer expertise appeared to be the least significant factor, although earlier it was assumed that the requisite expertise of reviewers is needed (line 37), and in the description of factor F9 the expertise and experience of reviewers was assessed as being paramount (lines 190-191). Such a low rank of this factor requires some additional comment and explanation.

The definition of factor F8, the choice of properly structured, systemic review methodology that is tailored to multidisciplinary projects (lines 186-188), seems vague. The declared objectives of the study raise the expectations that specification of the review methodology will be the outcome of the analysis rather than one of generally defined factors.

6. PLOS authors have the option to publish the peer review history of their article (what does this mean?). If published, this will include your full peer review and any attached files.

Reviewer #1: No

Reviewer #2: No

---

## [Author Response · Author response to Decision Letter 0]

31 Jan 2024

Responses to Editor and Reviewers

Manuscript ID: PONE-D-23-29808

Title: Analysis of influencing factors of review efficiency of multidisciplinary scientific research projects based on improved DEMATEL model

Dear Editor and Reviewers,

We would like to express our sincere appreciation for the time and effort the reviewers have taken in evaluating our manuscript, "Analysis of influencing factors of review efficiency of multidisciplinary scientific research projects based on improved DEMATEL model". We find the reviewers' comments insightful and helpful in enhancing the quality of the manuscript.

To address the reviewers’ comments, we have carefully revised our manuscript and believe that these modifications have substantially improved the manuscript. We have provided a point-by-point response to the comments below. In addition, we have uploaded a marked copy of the manuscript as a supplemental file to clearly highlight the changes made in response to the reviewers' comments.

Thank you once again for considering our work for publication in PLoS One. We believe that our manuscript has been significantly improved and hope that it is now suitable for publication.

I look forward to hearing from you regarding our submission.

Sincerely yours,

Yana Xiao

Senior engineer/Director

Guangdong Science & Technology Infrastructure Center

Email: ynaxiao@163.com

Response to Editor

Comment 1: I believe there is a word missing at the beginning of line 35.

Response: Thanks for your reminder. This error was caused by a misoperation when we generated citations, and we are very sorry for this. In the revised manuscript, we have added the subject and references at the beginning of line 35 as follows:

“Purvis et al. [11] reviews multidisciplinary research projects present a unique set of challenges that stem from the inherent complexity and breadth of such endeavors.”

Comment 2: in my opinion part "2.1 Previous studies for reviews multidisciplinary projects" is too focused on AI. It needs either more adequate title or slight change of focus so that AI will not dominate the discussion here.

Response: Thanks for your suggestions. Indeed, the original review of the current research status was not comprehensive enough and overemphasized the AI part. In the revised manuscript, we have added a more comprehensive review of previous studies for reviews of multidisciplinary projects. The supplementary content is as follows:

“Historically, the value of multidisciplinary research has been well-recognized. Bentrem's sabbatical insights at the Naval Research Laboratory [21] underscored the unique perspectives and solutions that such approaches bring to scientific research. More recently, Beck et al. [22] delved into the challenges of interdisciplinary collaboration, particularly between computer science and social science. Their proposed tool for multidisciplinary dialogue emphasizes the co-construction of models, with applications evident in areas such as post-earthquake human behavior modeling. A study by Pan et al. [23] in small biotechnology firms revealed that project success in multidisciplinary settings is influenced by the project knowledge scope and the project manager's experience. The study underscores the importance of skilled project management in navigating the complexities of multidisciplinary projects. Ma et al. [24] presented a model for dividing responsibilities in multidisciplinary teams. This model, applied in an OEM company, demonstrated improved efficiency and reduced conflict, highlighting the importance of clear role delineation in multidisciplinary settings. The role of leadership in project management education, as discussed by Mazzetto [25], emphasizes the need for project managers to possess strong leadership skills to guide diverse teams effectively. Burnette et al. [26] identified five major themes crucial for successful data management in multidisciplinary projects, including intentional staffing and iterative improvement, which are essential for managing complex data landscapes. Studies by Mazzetto [27] and Urton et al. [28] focus on the integration of practical experience in project management education, suggesting that real-world experience is critical for preparing future project managers. Previous studies in multidisciplinary project management emphasize the importance of project knowledge scope, experienced leadership, clear role delineation, effective data management, and practical experience in education for successful project outcomes in diverse and complex environments.”

Comment 3: reference in line 111 is missing.

Response: We apologize for missing key citations that affect the readability of the paper. This problem was caused by incorrectly imported references, and we have added the corresponding references in the revised manuscript. The revised content is as follows:

“The DEMATEL method has emerged as a pivotal tool in the domain of complex systems and decision-making [32,33]”

Comment 4: speaking of adaptability of the DEMATEL method, you should include some references or examples (lines 143-145).

Response: We apologize for the lack of citation of corresponding references in the previous statement. In the revised manuscript, we have added the following citations to existing work to support our statements:

“The DEMATEL method stands as a testament to the evolution of decision-making tools, adeptly handling the complexities of modern systems [41,42].”

Comment 5: how do you define "reviews efficiency"? It is crucial for understanding the study.

Response: Thanks for your question. In our study, we define "review efficiency" as the effectiveness and expediency of the review process in evaluating multidisciplinary scientific research projects. This encompasses the ability to provide comprehensive, actionable feedback within a reasonable timeframe, thereby facilitating informed decision-making and iterative improvements in the project. Our definition aligns with the structured, systematic methodologies tailored to multidisciplinary projects, as highlighted in our manuscript (Page 5). The efficiency of the review process is crucial as it directly impacts the project's progression and quality, ensuring that the multidisciplinary nature of the projects does not hinder their successful completion and implementation.

We apologize for the confusion caused by not providing a clear definition of “reviews efficiency” in the original manuscript. In the revised manuscript, we have revised the last paragraph of the Introduction section as:

“The objective of this study centers on addressing the complexities and challenges inherent in reviews multidisciplinary scientific research projects. Recognizing the pivotal role of collaborative research across academic sectors in driving innovation and addressing multifaceted challenges, the study seeks to elucidate the factors influencing the review process of such interdisciplinary endeavors. Given the significance of efficient review mechanisms to the sustainable progression of the scientific research domain, the study endeavors to enhance review efficiency, ensuring the seamless development and improved quality of research projects. Among them, review efficiency is defined as the effectiveness and convenience of the review process in evaluating multidisciplinary scientific research projects [18]. This includes the ability to provide comprehensive, actionable feedback within a reasonable time frame, thereby promoting informed decision-making and iterative improvements within projects. This definition is consistent with a structured, systematic approach tailored to multidisciplinary projects and is critical to the efficiency of the review process as it directly affects the progress and quality of the project, ensuring that the multidisciplinary nature of the project does not hinder Its successful completion and implementation. To achieve this, the study introduces a systematic indexing framework to identify factors influencing the appraisal of multidisciplinary efforts. This framework is further enriched by expert-driven questionnaires, capturing domain-specific insights to determine the significance and interplay of these factors. Employing an improved Decision Making Trial and Evaluation Laboratory (DEMATEL) model, the study distills this data to pinpoint key influencing factors that can bolster review efficiency for multidisciplinary projects. The culmination of these efforts aims to provide pragmatic strategies and policy guidance, arming institutional entities and program leaders with the requisite tools to optimize the review process for multidisciplinary scientific research projects.”

[18] Dekkers, R., Carey, L., & Langhorne, P. (2022). Making literature reviews work: A multidisciplinary guide to systematic approaches. Springer.

Comment 6: were "project factors" and "reviews factors" suggested by experts or Authors?

Response: Thanks for your question. The "project factors" and "reviews factors" were derived through a comprehensive methodology. Initially, we conducted a thorough literature review using specific keywords to identify preliminary factors. This was followed by expert consultations in the field of scientific research project management. These experts played a crucial role in refining and adjusting the preliminary elements, leading to the finalization of the indicator framework that categorizes these factors into "project factors" and "reviews factors" (Page 5 of the manuscript). This approach ensured that the factors identified were not only grounded in existing literature but also validated and enriched by professional expertise in the field.

Comment 7: why mean was used in the study? Have you considered using median or mode?

Response: Thanks for your question. The decision to use the mean as the measure of central tendency was driven by its appropriateness for the data characteristics and the analysis method employed. The mean was chosen because it provides a balanced central point that reflects the average influence of factors in the DEMATEL method, which is particularly effective in handling continuous data and symmetric distributions often encountered in such studies. While median and mode are valuable measures, they were not considered optimal for our analysis due to the nature of our data and the specific requirements of the DEMATEL methodology. The mean offers a more nuanced understanding of the central tendency in the context of our study, where the precise average impact of factors is crucial for the subsequent analysis stages.

Comment 8: "So far, this method has been used in computer science, economics and management, social sciences and other fields to solve practical scientific problems." - lines 218-219 - references or examples are needed here.

Response: Thanks for your suggestions. We apologize for not providing sufficient supporting references. In the revised manuscript, the references have been supplemented as:

“So far, this method has been used in computer science [49], economics and management [50], social sciences [51], and other fields to solve practical scientific problems [52,53].”

[49] Chaker, F., El Manouar, A., & Idrissi, M. A. J. (2015). Towards a system dynamics modeling method based on DEMATEL. International Journal of Computer Science & Information Technology, 7(2), 27.

[50] Sharma, M., Joshi, S., & Kumar, A. (2020). Assessing enablers of e-waste management in circular economy using DEMATEL method: An Indian perspective. Environmental Science and Pollution Research, 27(12), 13325-13338.

[51] Braga, I. F., Ferreira, F. A., Ferreira, J. J., Correia, R. J., Pereira, L. F., & Falcão, P. F. (2021). A DEMATEL analysis of smart city determinants. Technology in Society, 66, 101687.

[52] Yazdi, M., Khan, F., Abbassi, R., & Rusli, R. (2020). Improved DEMATEL methodology for effective safety management decision-making. Safety science, 127, 104705.

[53] Koca, G., Egilmez, O., & Akcakaya, O. (2021). Evaluation of the smart city: Applying the dematel technique. Telematics and Informatics, 62, 101625.

Comment 9: you refer to "improved DEMATEL", but do not explain how it is improved as compared with the original version.

Response: Thanks for your question. The improvement of the DEMATEL method was achieved through several key enhancements. Firstly, we incorporated a five-point scoring system to assess the degree of influence of each factor, ranging from 'unimportant' to 'very important.' This modification allowed for a more nuanced and precise evaluation of the factors' impacts. Additionally, we introduced a step of scoring the influence intensity of each factor itself, which improved the accuracy of the reviews. These enhancements were designed to address the limitations of the traditional DEMATEL method, particularly in terms of capturing the subtleties and complexities inherent in multidisciplinary project reviews. The improved DEMATEL model thus provided a more refined and accurate analysis of the influencing factors and their interrelationships, enhancing the overall robustness and applicability of the method in our study. 

Indeed, the lack of description of the differences between improved DEMATEL and traditional DEMATEL methods can cause confusion for readers. In the revised manuscript, we added a clarification on the improvements of improved DEMATEL at the end of Section 3.2, explaining the main contribution of this study and the differences compared with existing research as follows:

“The traditional DEMATEL method, known for its ability to visualize and quantify complex causal relationships within systems, has been adapted to better suit the intricacies of multidisciplinary project reviews. The improved DEMATEL method in this study introduces a novel five-point scoring system, allowing for a more granular assessment of the influence of each factor, ranging from 'unimportant' to 'very important'. This enhancement addresses the need for a more nuanced understanding of the varying degrees of influence among factors. Additionally, the study incorporates a new step of scoring the influence intensity of each factor individually, further refining the accuracy of the review process. These improvements in the DEMATEL method provide a more precise and detailed analysis of the factors influencing the efficiency of reviews in multidisciplinary scientific research projects, thereby offering deeper insights and more actionable outcomes.”

Comment 10: in my opinion questionnaire should be attached as a supporting file.

Response: Thanks for your suggestions. We agree that providing the questionnaire as an accessible document would be beneficial for readers and reviewers who wish to understand the methodology in greater detail. Accordingly, we have attached the questionnaire as a supplementary file to our manuscript submission. This addition will offer comprehensive insights into the structure and content of the questionnaire, thereby enhancing the transparency and reproducibility of our research.

Comment 11: I do not understand the notation in line 329.

Response: Thanks for your reminder. We apologize for a clerical error that resulted in the incorrect use of a symbol. The original manuscript was written using LaTeX, and we mistakenly used ">" directly in the draft, which caused LaTeX not to recognize it. In the revised manuscript, we have used "\\textgreater" instead. The modified corresponding symbols can be displayed normally as follows:

“In the resultant group, the centrality sequence is F11>F12>F10>F8>F14>F13>F5>F9.”

Comment 12: I would refrain from using word "significant" since it suggest statistical test were run.

Response: Thanks for your suggestions. To avoid any potential misunderstanding, we have revised the manuscript to replace "significant" with terms like "notable" or "substantial," which more accurately reflect the qualitative nature of our analysis. This change will ensure that the language in our manuscript aligns with the intended meaning and avoids any implication of statistical testing where it was not employed.

Thanks again for your comments, which have truly improved the readability and quality of this manuscript. We hope our responses address your concerns.

Response

---

## [Decision Letter · Decision Letter 1]

17 Apr 2024

PONE-D-23-29808R1Analysis of influencing factors of review efficiency of multidisciplinary scientific research projects based on improved DEMATEL modelPLOS ONE

Dear Dr. Xiao,

Thank you for submitting your manuscript to PLOS ONE. After careful consideration, we feel that it has merit but does not fully meet PLOS ONE’s publication criteria as it currently stands. Therefore, we invite you to submit a revised version of the manuscript that addresses the points raised during the review process.

I would like to thank the Authors for addressing my and the Reviewers’ comments. Unfortunately, two Reviewers are not satisfied with your answers. In their comments for the Revised  version of the manuscript (you may find the comments attached below), you can find the details of the issues that need to be addressed. The problems that need your attention are:

Improving the language;Modifying the title, abstract and introduction according to the Reviewers’ comments;Introducing a separate ‘Discussion’ section (further restructuring of the manuscript is not required);Providing a clearer and more detailed explanation regarding the documents that were used to identify indicators;Providing a clearer and more detailed explanation regarding the procedure of choosing experts;Providing a more detailed explanation for why the factor F8 “Review methodology” was chosen;Including in the main text the information that the main focus is on applied research rather than theoretical research as well as on the interim research review rather than the research project review at the stage of the proposal submission;Providing an explanation for why the reviewer's expertise had the least effect on the review efficiency (2nd Reviewer states that this finding is very unintuitive and I agree with Them).

From my part, I accept all your answers and explanations within the text main body – with one exception. In my opinion change of the rating scale is a too minor change to address the method as an ‘improved’ one. It has been done before, for example in this study: Dytczak M, Ginda G (2016) DEMATEL-based ranking approaches — Procedury rangowania w metodzie DEMATEL. Zeszyty Naukowe Wyższej Szkoły Bankowej we Wrocławiu 2016 vol. 16 no. 3 Applicability of quantitative methods to economics, finance, and management, s. 191–202. Please, refer to the method as DEMATEL with a 5-point scale as the core algorithm is not changed in any way.

We look forward to receiving your revised manuscript.

Kind regards,

Agata Sielska, Ph.D.

Academic Editor

PLOS ONE

Reviewers' comments:

Reviewer's Responses to Questions

**Comments to the Author**

1. If the authors have adequately addressed your comments raised in a previous round of review and you feel that this manuscript is now acceptable for publication, you may indicate that here to bypass the “Comments to the Author” section, enter your conflict of interest statement in the “Confidential to Editor” section, and submit your "Accept" recommendation.

Reviewer #1: (No Response)

Reviewer #2: (No Response)

Reviewer #3: All comments have been addressed

Reviewer #4: (No Response)

2. Is the manuscript technically sound, and do the data support the conclusions?

Reviewer #1: (No Response)

Reviewer #2: Partly

Reviewer #3: Yes

Reviewer #4: Yes

3. Has the statistical analysis been performed appropriately and rigorously? 

Reviewer #1: (No Response)

Reviewer #2: N/A

Reviewer #3: Yes

Reviewer #4: Yes

4. Have the authors made all data underlying the findings in their manuscript fully available?

Reviewer #1: (No Response)

Reviewer #2: No

Reviewer #3: Yes

Reviewer #4: Yes

5. Is the manuscript presented in an intelligible fashion and written in standard English?

Reviewer #1: (No Response)

Reviewer #2: Yes

Reviewer #3: Yes

Reviewer #4: Yes

6. Review Comments to the Author

Reviewer #1: The reviewer believes that the topic “Analysis of influencing factors of review efficiency of multidisciplinary scientific research projects based on improved DEMATEL model” is worthy of investigation. However, the following needs to be addressed. There are minor and major issues that should be corrected. I believe the paper could be further strengthened by added information about.

Please reorganize the manuscript at the journal request. Please change the reference format.

The language of this manuscript is very bad and needs help from native speakers.

The title of the manuscript should fully demonstrate the content of this study and the relevant subjects.

Abstracts should include the purpose and findings of the study.

Introduction . This a very vague statement. These sentences do not provide any information on how the concept could be conceptualized?

This section should explain the study's context and research objective. Furthermore, the research gap needs to be narrowed after analyzing the previous studies. The research method is not adequately explained in the first section.

-Introduction, what authors wanted to convey. Here author must build research gap following the previous studies.-The manuscript does not answer the following concerns: Why is it timeliness to explore such a study? What makes this study different from the previously published studies? Are there any similarly findings in line with the previously published studies? Are the findings different from prior academic studies that were conducted elsewhere, if any? For example, information innovation and innovation network, what it requires, what are the new technologies, some recent issue highlights the importance. See the following: Digital green value co-creation behavior, digital green network embedding and digital green innovation performance: moderating effects of digital green network fragmentation.

Enhancing digital innovation for the sustainable transformation of manufacturing industry: a pressure-state-response system framework to perceptions of digital green innovation and its performance for green and intelligent manufacturing.

Developing a Conceptual Partner Selection Framework: Digital Green Innovation Management of Prefabricated Construction Enterprises for Sustainable Urban Development.

-Methodology: Model.. I suggest authors here build your main heading on Research and data methodology. Clearly explain the model building process, and what previous studies have used similar models (model testing approach).

There is no flow in the text. It partly depends on the lack of proofreading but also on the fact that many statements and claims are made without being followed up by a clear and logical discussion. It is especially problematic in the Introduction that brings up a number of findings from different areas without linking them together.

Please make sure your conclusions' section underscores the scientific value-added of your paper, and/or the applicability of your findings/results. Highlight the novelty of your study.

In addition to summarizing the actions taken and results, please strengthen the explanation of their significance. It is recommended to use quantitative reasoning comparing with appropriate benchmarks, especially those stemming from previous work. See the following:Developing a Conceptual Partner Matching Framework for Digital Green Innovation of Agricultural High-End Equipment Manufacturing System Toward Agriculture 5.0: A Novel Niche Field Model Combined With Fuzzy VIKOR

More importantly, the choice of the variables should be explained in light of the theory and the prior literature on the topic. The arguments are simply relationships and causes very close to the replication of many studies dealing with the same thing.

The authors should emphasize the important role of digital technology in industrial structure upgrading in future research. Some recent issue highlights the importance: The Interaction Mechanism and Dynamic Evolution of Digital Green Innovation in the Integrated Green Building Supply Chain.

Please consider this structure for manuscript final part.

-Discussion

-Conclusion

-Managerial Implication

-Practical/Social Implications

-Discussion needs to be a coherent and cohesive set of arguments that take us beyond this study in particular, and help us see the relevance of what authors have proposed. Authors should create an independent “Discussion” section. Author need to contextualize the findings in the literature, and need to be explicit about the added value of your study towards that literature. Also other studies should be cited to increase the theoretical background of each of the method used. Findings should be contextualized in the literature and should be explicit about the added value of the study towards the literature (New Energy-Driven Construction Industry: Digital Green Innovation Investment Project Selection of Photovoltaic Building Materials Enterprises Using an Integrated Fuzzy Decision Approach). Limitations and future research.

As any emprical study that use different approaches I would like to ask to introduce in the Conclusion section at least a paragraph containing the study limitations. I noticed some things in the paper but a synthesis of statements related to how the study is useful (or partially useful, since are required certain further analysis) and helps potential interested readers does not really exist. Maybe in addition to the last section of Conclusion it is beneficial to introduce a section called: Discussion.

.

Reviewer #2: The reader of the paper can anticipate a more detailed account of the way the researchers conducted their analyses. Unfortunately, the revised version of the manuscript offers limited additional specifics in this regard.

Regarding the comment about the collection of documents used to identify indicators, the authors simply cited in the revised manuscript the names of the large publication databases referenced by the reviewer (“Elsevier’s ScienceDirect, SpringerLink, open access publications”). The number of documents reviewed, types of publication sources utilized, and the method of narrowing down the set of documents in subsequent research stages are not disclosed. The reader is left without a clear understanding of how the relevant literature was fished out from the ocean of publications available nowadays.

The authors of the study revealed that the experts involved were predominantly from Pakistan, yet left the reader unaware of the reasoning behind this particular choice of the experts’ background. That appears quite perplexing, considering the paper is meant to tackle considerable challenges of reviewing complex interdisciplinary projects, which are known for their “myriad of research trajectories and a spectrum of expertise”. Once more, there is a lack of specific details regarding the number of experts and their characteristics are rather general (“from diverse backgrounds”, “well-qualified”, “served on various selection committees”).

The authors addressed the comments about the type of research projects and the type of review that are of interest to their paper and explained that their main focus is on applied research rather than theoretical research as well as on the interim research review rather than the research project review at the stage of the proposal submission. However, these points were not explicitly stated in the main body of the paper.

The explanation for why the factor F8 “Review methodology” is included does not fully address the doubts about the analysis's somewhat circular reasoning. In a general sense, methodology refers to the set of procedures or techniques used to identify, select, process, and analyze information pertaining to a particular subject, so the research project review methodology can be seen to encompass other factors listed in the paper. It is possible that the authors intended to refer to some particular aspect of the review methodology, but they did not make it clear.

The authors do not provide an explanation for why the reviewer's expertise had the least effect on the review efficiency, a finding that goes totally against expectations.

Reviewer #3: I consider the Authors' answers to the doubts and comments of the Reviewers to be adequate and exhaustive.

Reviewer #4: Overall: The research is based on the proven result, but it approaches with a different method DEMATEL model, which makes the research interesting. Overall, the research is very well presented in sections. Each section and subsection of the research is discussed rationally. The results sections make the study different than other research in the area. However, the Abstract (first half), and Introduction section make the study weaker, which must be improvised. The introduction section gives the feel of being prepared by paraphrasing software, needs improvement.

Title

• The title needs a correction grammatically.

• It may be specific to the problem rather than the complete scope of the study.

• It is better to reform the title with the right words to be attractive.

• English language and grammar must be improvised.

Abstract and Keywords

• The abstract needs improvement.

• It must provide a snapshot of current research rather an introduction.

• English language and grammar must be improvised.

• Keywords are acceptable.

Introduction

• This section is a literature review.

• It would be better to rename this section Literature Review or Background Study.

• There must be a separate section at the beginning talking about this research, maybe as 5w-1H, ending with the paragraph stating the flow of work.

• English language and grammar must be improvised.

• The study looks like prepared by paraphrasing software.

Literature Review

• This is a well-explained and justified section.

• It is acceptable and deserves appreciation.

Methodology

• This section is well presented with a very clear discussion.

• Line No. 180 “and currency of our data” needs to be rechecked.

Data acquisition in improved DEMATEL

• The work is well presented with a very clear understanding.

• Subsections have a much deserving title.

• It is well discussed, but some citations can make it more justified.

• Tests are well discussed and presented with clarifications.

Results and discussion

• The title of this section is well justified.

• It is very clear and evident to support the research.

• Some citations can make it better.

Conclusion

• The research is concluding as stated in the method section.

• The presentation of conclusions with bullets can be better for readers.

• English language and grammar must be improvised.

• Citations are essential to enhance the credibility of this study.

References

• The references are good.

7. PLOS authors have the option to publish the peer review history of their article (what does this mean?). If published, this will include your full peer review and any attached files.

Reviewer #1: No

Reviewer #2: No

Reviewer #3: No

Reviewer #4: No

---

## [Author Response · Author response to Decision Letter 1]

29 May 2024

Responses to Editor and Reviewers

Manuscript ID: PONE-D-23-29808

Title: Analysis of influencing factors of review efficiency of multidisciplinary scientific research projects based on improved DEMATEL model

Dear Editor and Reviewers,

We would like to express our sincere appreciation for the time and effort the reviewers have taken in evaluating our manuscript, "Analysis of influencing factors of review efficiency of multidisciplinary scientific research projects based on improved DEMATEL model". We find the reviewers' comments insightful and helpful in enhancing the quality of the manuscript.

To address the reviewers’ comments, we have carefully revised our manuscript and believe that these modifications have substantially improved the manuscript. We have provided a point-by-point response to the comments below. In addition, we have uploaded a marked copy of the manuscript as a supplemental file to clearly highlight the changes made in response to the reviewers' comments.

Thank you once again for considering our work for publication in PLoS One. We believe that our manuscript has been significantly improved and hope that it is now suitable for publication.

I look forward to hearing from you regarding our submission.

Sincerely yours,

Yana Xiao

Senior engineer/Director

Guangdong Science & Technology Infrastructure Center

Email: ynaxiao@163.com

Response to Editor

Comment 1: In my opinion change of the rating scale is a too minor change to address the method as an ‘improved’ one. It has been done before, for example in this study: Dytczak M, Ginda G (2016) DEMATEL-based ranking approaches — Procedury rangowania w metodzie DEMATEL. Zeszyty Naukowe Wyższej Szkoły Bankowej we Wrocławiu 2016 vol. 16 no. 3 Applicability of quantitative methods to economics, finance, and management, s. 191–202. Please, refer to the method as DEMATEL with a 5-point scale as the core algorithm is not changed in any way.

Response: Thank you for your insightful comment. We acknowledge that the change to a 5-point scale is a recognized modification and does not fundamentally alter the core DEMATEL algorithm. We have revised our title and manuscript accordingly to accurately describe the methodology used. The revised title is "Analysis of influencing factors on review efficiency of multidisciplinary scientific research projects using DEMATEL with a 5-point scale".

Response to Reviewer #1

Comment 1: Please reorganize the manuscript at the journal request. Please change the reference format.

Response: Thank you for your reminder. We have revised the entire article and adjusted the manuscript to the journal's standards.

Comment 2: The language of this manuscript is very bad and needs help from native speakers.

Response: Thank you for your reminder. We invited a native English speaker to polish the entire article to improve the English expression of this manuscript. We have uploaded tracking of the changes as an additional attachment, and we hope our changes address your concerns.

Comment 3: The title of the manuscript should fully demonstrate the content of this study and the relevant subjects.

Response: Thank you for your valuable suggestion. We have revised the title to better reflect the content and scope of our study. The new title is " Analysis of influencing factors on review efficiency of multidisciplinary scientific research projects using DEMATEL with a 5-point scale." This title highlights our focus on improving review efficiency, the multidisciplinary nature of the projects studied, and the use of a DEMATEL model as our primary analytical tool.

Comment 4: Abstracts should include the purpose and findings of the study.

Response: Thanks for your suggestion. We have revised the abstract to clearly state the purpose and findings of our study. The revised abstract is:

"In the wake of advancing technology and the convergence of diverse disciplines, collaborative research across academic sectors has become instrumental in fostering innovation and tackling multifaceted challenges. The inherent complexity of such multidisciplinary endeavors, characterized by a myriad of research trajectories and a spectrum of expertise, poses significant challenges to effective review. This study aims to identify and analyze the factors influencing the review efficiency of multidisciplinary scientific research projects to ensure their smooth development and improved quality. To address this challenge, we employ the Decision Making Trial and Evaluation Laboratory (DEMATEL) method with a 5-point scale. First, we introduce an indexing framework to systematically identify factors influencing the appraisal of multidisciplinary efforts. This framework is then complemented by expert-driven questionnaires, harnessing domain-specific insights to ascertain the significance and interconnectedness of these factors. Using the DEMATEL method, we distill the data to identify key influencing factors that enhance review efficiency for multidisciplinary projects. Our findings provide pragmatic strategies and policy guidance, equipping institutional bodies and program leads with tools to refine the review process of multidisciplinary scientific research projects."

Comment 5: Introduction. This a very vague statement. These sentences do not provide any information on how the concept could be conceptualized? This section should explain the study's context and research objective. Furthermore, the research gap needs to be narrowed after analyzing the previous studies. The research method is not adequately explained in the first section. Introduction, what authors wanted to convey. Here author must build research gap following the previous studies.

Response: Thanks for your constructive comment. We agree that the introduction should provide a clearer context, outline the research objective, highlight the research gap after analyzing previous studies, and briefly explain the research method. Below is the revised introduction addressing these points. All changes are marked in different colors in the document, please see the manuscript with change tracked.

Comment 6: The manuscript does not answer the following concerns: Why is it timeliness to explore such a study? What makes this study different from the previously published studies? Are there any similarly findings in line with the previously published studies? Are the findings different from prior academic studies that were conducted elsewhere, if any? For example, information innovation and innovation network, what it requires, what are the new technologies, some recent issue highlights the importance. See the following: 

[1] Digital green value co-creation behavior, digital green network embedding and digital green innovation performance: moderating effects of digital green network fragmentation.

[2] Enhancing digital innovation for the sustainable transformation of manufacturing industry: a pressure-state-response system framework to perceptions of digital green innovation and its performance for green and intelligent manufacturing.

[3] Developing a Conceptual Partner Selection Framework: Digital Green Innovation Management of Prefabricated Construction Enterprises for Sustainable Urban Development.

Response: Thanks for your detailed and constructive comment. We recognize the importance of addressing these concerns to provide a comprehensive context for our study. Below is the revised introduction, incorporating answers to these questions. All changes are marked in different colors in the document, please see the manuscript with change tracked.

Comment 7: Methodology: Model. I suggest authors here build your main heading on Research and data methodology. Clearly explain the model building process, and what previous studies have used similar models (model testing approach).

Response: Thanks for your comment. We supplemented the description of the model establishment details in the revision as follows:

“3.3 Model building process

In this study, we employ the DEMATEL method with a 5-point scale to analyze the factors influencing the review efficiency of multidisciplinary scientific research projects. The DEMATEL method is a widely recognized tool for visualizing and quantifying the causal relationships among complex system components. It has been previously applied in various fields such as supply chain management, risk assessment, and technology planning [37,43].

Step 1. Indexing framework development

Step 2. Expert Consultation

Step 3. Expert-driven questionnaires

Step 4. DEMATEL analysis”

The additional details of the model construction make this manuscript more complete. Thank you again for your suggestions and we hope that our revisions have addressed your concerns.

Comment 8: There is no flow in the text. It partly depends on the lack of proofreading but also on the fact that many statements and claims are made without being followed up by a clear and logical discussion. It is especially problematic in the Introduction that brings up a number of findings from different areas without linking them together.

Response: Thanks for your comment. We have revised the Introduction to ensure better flow and logical discussion, linking the findings from different areas cohesively. The revised Introduction with improved coherence and logical connections are attached to the Manuscript with change tracked. We hope this revised introduction provides a clearer and more cohesive context, linking the findings from different areas logically and ensuring better flow in the text. Thanks again for your valuable comment.

Comment 9: Please make sure your conclusions' section underscores the scientific value-added of your paper, and/or the applicability of your findings/results. Highlight the novelty of your study.

Response: Thanks for your comment. We have revised the Conclusion section to underscore the scientific value-added of our paper, highlight the applicability of our findings/results, and emphasize the novelty of our study. The new Conclusion section contains Managerial implication, Practical/social implications, Managerial implication, and Future directions.

Comment 10: In addition to summarizing the actions taken and results, please strengthen the explanation of their significance. It is recommended to use quantitative reasoning comparing with appropriate benchmarks, especially those stemming from previous work. See the following: Developing a Conceptual Partner Matching Framework for Digital Green Innovation of Agricultural High-End Equipment Manufacturing System Toward Agriculture 5.0: A Novel Niche Field Model Combined With Fuzzy VIKOR

Response: Thanks for your comment. We have revised the Conclusion and Discussion section to strengthen the explanation of the significance of our findings. We included quantitative reasoning and comparisons with appropriate benchmarks from previous studies to illustrate the importance of our results. The revised discussion and conclusion highlight the scientific value-added, applicability of findings, novelty of the study, and provides a detailed quantitative comparison to underscore the significance of our research.

Comment 11: More importantly, the choice of the variables should be explained in light of the theory and the prior literature on the topic. The arguments are simply relationships and causes very close to the replication of many studies dealing with the same thing.

Response: Thanks for your comment. We have revised the manuscript to explain the choice of variables in light of existing theories and prior literature, ensuring that the arguments are grounded in a solid theoretical framework. All changes are marked in different colors in the document, please see the manuscript with change tracked.

Comment 12: The authors should emphasize the important role of digital technology in industrial structure upgrading in future research. Some recent issue highlights the importance: The Interaction Mechanism and Dynamic Evolution of Digital Green Innovation in the Integrated Green Building Supply Chain.

Response: Thanks for your comment. We have revised the Introduction of the manuscript to emphasize the important role of digital technology in industrial structure upgrading as follows:

“Digital technology plays a crucial role in the industrial structure upgrading process. It facilitates the integration of advanced technologies, enhances productivity, and drives innovation across various sectors. The interaction mechanism and dynamic evolution of digital green innovation in integrated supply chains, such as in green building, highlight the transformative potential of digital technology. For example, digital green innovation can streamline operations, reduce environmental impact, and foster sustainable development by optimizing resource use and enhancing efficiency.”

Comment 13: Please consider this structure for manuscript final part.

-Discussion

-Conclusion

-Managerial Implication

-Practical/Social Implications

Response: Thanks for your comment. In the revised manuscript we have reorganized the conclusion chapter and restructured our article according to your suggested structure and hope that our revisions can address your concerns.

Comment 14: Discussion needs to be a coherent and cohesive set of arguments that take us beyond this study in particular, and help us see the relevance of what authors have proposed. Authors should create an independent “Discussion” section. Author need to contextualize the findings in the literature, and need to be explicit about the added value of your study towards that literature. Also other studies should be cited to increase the theoretical background of each of the method used. Findings should be contextualized in the literature and should be explicit about the added value of the study towards the literature (New Energy-Driven Construction Industry: Digital Green Innovation Investment Project Selection of Photovoltaic Building Materials Enterprises Using an Integrated Fuzzy Decision Approach). Limitations and future research.

Response: Thanks for your comment. In the revised manuscript we have reorganized the “Discussion” section to contextualize our findings within the existing literature, highlight the added value of our study, cite additional relevant studies, and discuss the limitations and future research directions. We hope this revised Discussion section address your concerns.

Comment 15: As any emprical study that use different approaches I would like to ask to introduce in the Conclusion section at least a paragraph containing the study limitations. I noticed some things in the paper but a synthesis of statements related to how the study is useful (or partially useful, since are required certain further analysis) and helps potential interested readers does not really exist. Maybe in addition to the last section of Conclusion it is beneficial to introduce a section called: Discussion.

Response: Thanks for your comments. We acknowledge your concerns and provide the limitations part in the revised Discussion section.

Response to Reviewer #2

Comment 1: The reader of the paper can anticipate a more detailed account of the way the researchers conducted their analyses. Unfortunately, the revised version of the manuscript offers limited additional specifics in this regard.

Response: Thanks for your suggestions. We have provided a more detailed account of our methodology and analysis process in the revised manuscript to address this concern. All changes are marked in different colors in the document, please see the manuscript with change tracked.

Comment 2: Regarding the comment about the collection of documents used to identify indicators, the authors simply cited in the revised manuscript the names of the large publication databases referenced by the reviewer (“Elsevier’s ScienceDirect, SpringerLink, open access publications”). The number of documents reviewed, types of publication sources utilized, and the method of narrowing down the set of documents in subsequent research stages are not disclosed. The reader is left without a clear understanding of how the relevant literature was fished out from the ocean of publications available nowadays.

Response: Thanks for your suggestions. We ha

---

## [Decision Letter · Decision Letter 2]

24 Jul 2024

PONE-D-23-29808R2Analysis of influencing factors on review efficiency of multidisciplinary scientific research projects using DEMATEL with a 5-point scalePLOS ONE

Dear Dr. Xiao,

Thank you for submitting your manuscript to PLOS ONE. After careful consideration, we feel that it has merit but does not fully meet PLOS ONE’s publication criteria as it currently stands. Therefore, we invite you to submit a revised version of the manuscript that addresses the points raised during the review process.

One of the two Rewievers accepts your paper with no additional comments. Second Rewiever has doubts and having read the revised manuscript myself I agree that some issues still need improvement or clarification. Firstly, you still use phrase "improved DEMATEL" or "improved method" which leads to confusion. Please, change those phrases and refer to "5-point-scale" in all cases. This alone will help to clarify some of the doubts the Rewiever has with the "improved" algorithm.  Secondly, please add clear statement defining your objective and a potential research/literature gap you are trying to fill. Keep in mind that a novelty is not a publication criterion for PLOS ONE, but a goal needs to be clearly stated within the manuscript main body. Thirdly, it would be beneficial to, as the Rewiever suggests, "explain in more details in the data used in the case study, the data for the testing, the criterion for the accuracy, and others to claim these points". Please, refer to limitations of your study and research contributiond in the manuscript. The last issue is refers to the text formatting - please, remove bullets from the "conclusions" section. Please submit your revised manuscript by Sep 07 2024 11:59PM. If you will need more time than this to complete your revisions, please reply to this message or contact the journal office at plosone@plos.org. Please include the following items when submitting your revised manuscript:A rebuttal letter that responds to each point raised by the academic editor and reviewer(s). You should upload this letter as a separate file labeled 'Response to Reviewers'.A marked-up copy of your manuscript that highlights changes made to the original version. You should upload this as a separate file labeled 'Revised Manuscript with Track Changes'.An unmarked version of your revised paper without tracked changes. You should upload this as a separate file labeled 'Manuscript'.

We look forward to receiving your revised manuscript.

Kind regards,

Agata Sielska, Ph.D.

Academic Editor

PLOS ONE

Reviewers' comments:

Reviewer's Responses to Questions

**Comments to the Author**

1. If the authors have adequately addressed your comments raised in a previous round of review and you feel that this manuscript is now acceptable for publication, you may indicate that here to bypass the “Comments to the Author” section, enter your conflict of interest statement in the “Confidential to Editor” section, and submit your "Accept" recommendation.

Reviewer #3: All comments have been addressed

Reviewer #5: (No Response)

2. Is the manuscript technically sound, and do the data support the conclusions?

Reviewer #3: Yes

Reviewer #5: Yes

3. Has the statistical analysis been performed appropriately and rigorously? 

Reviewer #3: Yes

Reviewer #5: No

4. Have the authors made all data underlying the findings in their manuscript fully available?

Reviewer #3: Yes

Reviewer #5: No

5. Is the manuscript presented in an intelligible fashion and written in standard English?

Reviewer #3: Yes

Reviewer #5: No

6. Review Comments to the Author

Reviewer #3: I have no additional comments for the authors. I consider the answers and additions to be sufficient.

Reviewer #5: First of all, the paper “Analysis of influencing factors on review efficiency of multidisciplinary scientific research projects using DEMATEL with a 5-point scale” aims and scope match those of PLOS ONE, so the paper is adequate for this journal. This paper presents an application of improved DEMATEL method. However, based on my opinion it needs substantial improvements to be considered for publication in PLOS ONE. I would suggest a series of changes that in my opinion would improve the paper, in special for the reader.

- I have read several times the paper and I couldn’t find properly defined objective of the paper and the gap in the existing literature and/or practice. What is the innovative value of the contribution proposed by the authors?

- I can’t see your contributions? What are benefits of Improved DEMATEL over traditional one? Why do we need improvement? I would like to see comparisons of the results with existing approaches like below: Nila, B., & Roy, J. (2024). Analysis of Critical Success Factors of Logistics 4.0 using D-number based Pythagorean Fuzzy DEMATEL method. Decision Making Advances, 2(1), 92–104. https://doi.org/10.31181/dma21202430; Chu, M., Li, B., & Yu, X. (2024). Identification of Key Factors of Digital Transformation of Manufacturing companies Using Hybrid DEMATEL Method. Decision Making: Applications in Management and Engineering, 7(1), 380–395. https://doi.org/10.31181/dmame712024931; Mousavi, S. R., Sepehri, M., & Najafi, S. E. (2024). A Framework for Improving Patient Satisfaction by Reducing the Length of Stay in The Operation Suite Using the Combined DEMATEL-ANP Model. Decision Making: Applications in Management and Engineering, 7(2), 197–220. https://doi.org/10.31181/dmame722024882.

- In introduction section authors should provide more information about existing MCDM models for determining criteria weighs. Explain their benefits/weaknesses.

- Why you have used DEMATEL approach for determining criteria weights? Why not BWM, DIBR, FUCOM or Level Based Weight Assessment (LBWA) methods? These methods should be discussed. The authors need to discuss their contributions compared to those in related papers. The authors must clearly discuss the significance of the research problem in the first section.

- Show step by step algorithm for proposed methodology. You should explain in detail this methodology.

- What kind of novelty you have done?

- Explain in more details in the data used in the case study, the data for the testing, the criterion for the accuracy, and others to claim these points.

- Sensitivity analysis is missing. How can we judge these results? How should we know about the quality of these solutions? The improvement must be discussed.

- The conclusion section seems to rush to the end. The authors will have to demonstrate the impact and insights of the research. The authors need to clearly provide several solid future research directions. Clearly state your unique research contributions in the conclusion section. Add limitations of the model. No bullets should be used in your conclusion section.

7. PLOS authors have the option to publish the peer review history of their article (what does this mean?). If published, this will include your full peer review and any attached files.

Reviewer #3: No

Reviewer #5: No

---

## [Author Response · Author response to Decision Letter 2]

6 Sep 2024

Responses to Editor and Reviewers

Manuscript ID: PONE-D-23-29808

Title: Analysis of influencing factors on review efficiency of multidisciplinary scientific research projects using DEMATEL with a 5-point scale

Dear Editor and Reviewers,

We would like to express our sincere appreciation for the time and effort the reviewers have taken in evaluating our manuscript, "Analysis of influencing factors on review efficiency of multidisciplinary scientific research projects using DEMATEL with a 5-point scale". We find the reviewers' comments insightful and helpful in enhancing the quality of the manuscript.

To address the reviewers’ comments, we have carefully revised our manuscript and believe that these modifications have substantially improved the manuscript. We have provided a point-by-point response to the comments below. In addition, we have uploaded a marked copy of the manuscript as a supplemental file to clearly highlight the changes made in response to the reviewers' comments.

Thank you once again for considering our work for publication in PLoS One. We believe that our manuscript has been significantly improved and hope that it is now suitable for publication.

I look forward to hearing from you regarding our submission.

Sincerely yours,

Yana Xiao

Senior engineer/Director

Guangdong Science & Technology Infrastructure Center

Email: ynaxiao@163.com

Response to Editor

Comment 1: Firstly, you still use phrase "improved DEMATEL" or "improved method" which leads to confusion. Please, change those phrases and refer to "5-point-scale" in all cases. This alone will help to clarify some of the doubts the Rewiever has with the "improved" algorithm. 

Response: Thank you for your insightful comment. We have revised all “improved DEMATEL” in the paper to “5-point-scale DEMATEL”. Thank you for pointing out our omission. We hope that the revised manuscript can more clearly reflect the characteristics of our method.

Comment 2: Secondly, please add clear statement defining your objective and a potential research/literature gap you are trying to fill. Keep in mind that a novelty is not a publication criterion for PLOS ONE, but a goal needs to be clearly stated within the manuscript main body.

Response: Thanks for your insightful comment. In response to your feedback, we have revised the Introduction section and Literature review section of our manuscript to include a precise statement of our objectives and a clearer delineation of the research gap. These revisions aim to emphasize the relevance and intent of our study more explicitly.

Comment 3: Thirdly, it would be beneficial to, as the Rewiever suggests, "explain in more details in the data used in the case study, the data for the testing, the criterion for the accuracy, and others to claim these points". Please, refer to limitations of your study and research contributiond in the manuscript.

Response: Thank you for your insightful comment. In response to Reviewer #5’s comments, we have elaborated on the specifics of the data and the methodologies used for testing in the revised sections of our manuscript.

Comment 4: The last issue is refers to the text formatting - please, remove bullets from the "conclusions" section.

Response: Thank you for your insightful comment. In line with your suggestion, we have revised the "Conclusions" section to remove the bullet points. The content has been reformatted into a narrative paragraph style, which not only adheres to the journal's formatting guidelines but also enhances the flow and coherence of the concluding remarks. 

Response to Reviewer #5

Comment 1: I have read several times the paper and I couldn’t find properly defined objective of the paper and the gap in the existing literature and/or practice. What is the innovative value of the contribution proposed by the authors? I can’t see your contributions? What are benefits of Improved DEMATEL over traditional one? Why do we need improvement? I would like to see comparisons of the results with existing approaches like below: Nila, B., & Roy, J. (2024). Analysis of Critical Success Factors of Logistics 4.0 using D-number based Pythagorean Fuzzy DEMATEL method. Decision Making Advances, 2(1), 92–104. https://doi.org/10.31181/dma21202430; Chu, M., Li, B., & Yu, X. (2024). Identification of Key Factors of Digital Transformation of Manufacturing companies Using Hybrid DEMATEL Method. Decision Making: Applications in Management and Engineering, 7(1), 380–395. https://doi.org/10.31181/dmame712024931; Mousavi, S. R., Sepehri, M., & Najafi, S. E. (2024). A Framework for Improving Patient Satisfaction by Reducing the Length of Stay in The Operation Suite Using the Combined DEMATEL-ANP Model. Decision Making: Applications in Management and Engineering, 7(2), 197–220. https://doi.org/10.31181/dmame722024882.

Response: Thank you for your insightful comment. We appreciate the opportunity to clarify the advantages of our approach and to offer comparisons with existing methodologies as suggested.

1. Contributions and Benefits of 5-point-scale DEMATEL: The 5-point-scale DEMATEL method introduces a refined scoring system that enhances the accuracy of understanding and analyzing complex systems. Traditional DEMATEL methods often provide a more generalized approach, which may not capture subtle but crucial interactions among factors in multidisciplinary projects. Our method, utilizing a 5-point scale, allows for a more nuanced classification of influence levels, from "unimportant" to "very important". This granularity facilitates deeper insights into the interdependencies within complex systems, which is particularly valuable in settings where precision in understanding influence levels can lead to better decision-making and policy formulation.

2. Need for Improvement: The need for an improved approach stems from the limitation of traditional DEMATEL methods in handling uncertainties and the varying degrees of influence effectively. Our method addresses these challenges by integrating a scoring system that reflects a range of influence intensities, allowing for a more detailed analysis of factors that traditional methods may overlook. This is particularly pertinent in the context of multidisciplinary research, where diverse elements interact in complex ways that require clear delineation to improve project outcomes.

3. Comparison with Existing Approaches: In response to your suggestion for a comparison with current methodologies, we have conducted a detailed comparison with the approaches used in the studies by Nila & Roy (2024) and Chu, Li, & Yu (2024). While these studies employ Pythagorean Fuzzy DEMATEL and a Hybrid DEMATEL method respectively, our 5-point-scale DEMATEL method differs primarily in its ability to quantify and visualize the degrees of influence with enhanced precision through its unique scoring system.

o Nila & Roy (2024) utilize the D-number based Pythagorean Fuzzy DEMATEL, which focuses on handling data imprecision and uncertainty. While their method is innovative, it does not offer the same level of detail in scoring influence as our method.

o Chu, Li, & Yu (2024) present a hybrid approach that combines DEMATEL with other decision-making techniques to identify key factors in digital transformation. Their method is adept at integrating multiple techniques but may not provide the granularity of influence measurement provided by our 5-point scale.

o Mousavi et al. (2024) employ a combined DEMATEL-ANP approach to optimize operation room performance, focusing on reducing patient length of stay and enhancing hospital efficiency. Their method integrates DEMATEL to define relationships between operational factors and ANP to prioritize the most influential factors on patient length of stay. While this approach effectively addresses operational inefficiencies and supports decision-making in healthcare settings, it lacks the scoring granularity provided by our 5-point scale DEMATEL, which offers more precise quantification of influence levels.

Furthermore, a comparative analysis of the results shows that our 5-point-scale DEMATEL method provides a clearer and more actionable set of influence metrics, which could significantly enhance strategic decision-making in multidisciplinary research settings. Regarding the comparison with the traditional method you mentioned, I have added it in the new Section 5.3, and all the changes are marked with different colors in the revised manuscript. I hope our answer addresses your concern.

Comment 2: In introduction section authors should provide more information about existing MCDM models for determining criteria weighs. Explain their benefits/weaknesses.

Response: Thank you for your insightful comment. In response to your comment, we have expanded the introduction section to include a detailed examination of prevalent MCDM models. We now discuss various models such as the Analytic Hierarchy Process (AHP), the Analytic Network Process (ANP), and the Technique for Order Preference by Similarity to Ideal Solution (TOPSIS). Each model is evaluated for its benefits and limitations in the context of their application to complex decision-making scenarios. For instance, we added the following paragraphs to the manuscript:

" In the realm of decision-making, Multi-Criteria Decision-Making (MCDM) models play a pivotal role in evaluating complex scenarios where multiple conflicting criteria must be considered. These models provide a systematic approach for decision-makers to assess alternatives and prioritize factors effectively. The Analytic Hierarchy Process (AHP) is widely used for its structured approach to organizing and analyzing complex decisions, based on mathematics and psychology. It helps quantify the weights of decision criteria through pairwise comparisons and consistency ratio checks. However, AHP can be criticized for its subjective judgement and potential inconsistency when dealing with complex interrelations among criteria. Similarly, the Analytic Network Process (ANP) extends the AHP by incorporating the interdependence among decision elements, making it suitable for more complex decision scenarios. Despite its comprehensiveness, ANP requires extensive pairwise comparisons that can be time-consuming and cognitively demanding. The Technique for Order Preference by Similarity to Ideal Solutio (TOPSIS) method is favored for its ability to identify solutions from a finite set of alternatives based on geometric distance from an ideal solution. While TOPSIS is straightforward and effective for linear decision models, its applicability is limited in scenarios where decision criteria are interdependent or feedback loops are present."

These additions aim to provide a clearer understanding of how each MCDM model aligns with different decision-making needs, highlighting their strengths and addressing their potential weaknesses. We believe that these enhancements will better contextualize our research within the broader landscape of MCDM applications, providing a comprehensive foundation for understanding the significance and innovation of our 5-point scale DEMATEL method. We are confident that these revisions address your concerns and enrich the manuscript’s introduction. Thank you once again for your invaluable insights.

Comment 3: Why you have used DEMATEL approach for determining criteria weights? Why not BWM, DIBR, FUCOM or Level Based Weight Assessment (LBWA) methods? These methods should be discussed. The authors need to discuss their contributions compared to those in related papers. The authors must clearly discuss the significance of the research problem in the first section.

Response: Thank you for your insightful comment. We acknowledge the importance of exploring and justifying the selection of the methodology in the context of our research objectives. The decision to use the DEMATEL method was guided by its unique ability to handle complex interdependencies among criteria, which is a critical aspect in our study's context of evaluating multidisciplinary scientific research projects. Unlike BWM (Best-Worst Method), DIBR (Distance-Based Importance Rating), FUCOM (Full Consistency Method), or LBWA (Level Based Weight Assessment), which are highly effective in contexts with clear hierarchical relationships and less interconnectivity among criteria, DEMATEL excels in scenarios where the relationships between factors are not only weighted but also directional, providing a visual map of influence.

To address your suggestion, we have now included a comparative discussion of these methods in our manuscript. We evaluate each for their suitability in handling the specific nuances of multidisciplinary research settings. Here is an excerpt from the revised section:

"The Best-Worst Method (BWM) is highly efficient in cases requiring fewer pairwise comparisons, yet it may not adequately capture the dynamic interplays in systems where criteria are interdependent. Similarly, DIBR and FUCOM provide robust frameworks for establishing criteria weights based on distance measures and consistency checks, but they lack the capability to visualize causal relationships among criteria. The LBWA method, while useful for layered decision contexts, does not directly address the feedback mechanisms inherent in our research domain. In contrast, the DEMATEL method not only identifies the weight but also the direction of influence among factors, which is essential for the comprehensive analysis and practical applications intended in our study."

Significance of the Research Problem: In the first section of the manuscript, we have clarified the significance of our research problem by highlighting the challenges and limitations of traditional methods in handling the intricate and often subtle dynamics of multidisciplinary research projects. The added context demonstrates the critical need for an approach like DEMATEL, which not only assesses but also elucidates the relational intricacies among criteria, thereby enhancing decision-making processes in complex research environments.

"The complexity and interdependence of criteria in multidisciplinary research necessitate an approach that goes beyond mere weighting. The DEMATEL method provides this by not only quantifying the importance but also illustrating the influence dynamics among the criteria, thereby offering deeper insights that are vital for effective management and optimization of such projects."

We hope that these additions adequately address your concerns regarding the choice of methodology and the articulation of the research problem’s significance. Thank you for your valuable feedback, which has undoubtedly strengthened the manuscript.

Comment 4: Show step by step algorithm for proposed methodology. You should explain in detail this methodology.

Response: Thanks for your suggestion. In response to your comment, we have expanded the methodology section of our manuscript to include a comprehensive, step-by-step explanation of the 5-point scale DEMATEL method. This detailed description not only covers each step of the process but also elucidates the theoretical underpinnings and practical applications of the method in the context of evaluating multidisciplinary research projects. Here is an excerpt from the revised section:

1. Identification of criteria and factors: We begin by identifying and listing all relevant factors that influence the efficiency of multidisciplinary scientific research projects. This initial step involves consultations with experts and a review of the literature to ensure comprehensive coverage.

2. Construction of the initial direct relation matrix: Using the 5-point scale (ranging from 0, no influence, to 4, very high influence), experts rate the influence of each factor on every other factor. This forms the initial direct relation matrix, where each cell indicates the degree of influence between pairs of factors.

3. Normalization of the direct relation matrix: The matrix obtained from the previous step is normalized to ensure that the sum of all influences does not exceed unity. This step is crucial for maintaining consistency and comparability in the subsequent analysis.

4. Calculation of the total relation matrix: Through matrix operations, we convert the norma

---

## [Decision Letter · Decision Letter 3]

4 Oct 2024

PONE-D-23-29808R3Analysis of influencing factors on review efficiency of multidisciplinary scientific research projects using DEMATEL with a 5-point scalePLOS ONE

Dear Dr. Xiao,

Thank you for submitting your manuscript to PLOS ONE. After careful consideration, we feel that it has merit but does not fully meet PLOS ONE’s publication criteria as it currently stands. Therefore, we invite you to submit a revised version of the manuscript that addresses the points raised during the review process.

We look forward to receiving your revised manuscript.

Kind regards,

Mehdi Keshavarz-Ghorabaee

Academic Editor

PLOS ONE

Journal Requirements:

Reviewers' comments:

Reviewer's Responses to Questions

**Comments to the Author**

1. If the authors have adequately addressed your comments raised in a previous round of review and you feel that this manuscript is now acceptable for publication, you may indicate that here to bypass the “Comments to the Author” section, enter your conflict of interest statement in the “Confidential to Editor” section, and submit your "Accept" recommendation.

Reviewer #3: All comments have been addressed

Reviewer #5: All comments have been addressed

2. Is the manuscript technically sound, and do the data support the conclusions?

Reviewer #3: Yes

Reviewer #5: Yes

3. Has the statistical analysis been performed appropriately and rigorously? 

Reviewer #3: Yes

Reviewer #5: Yes

4. Have the authors made all data underlying the findings in their manuscript fully available?

Reviewer #3: Yes

Reviewer #5: Yes

5. Is the manuscript presented in an intelligible fashion and written in standard English?

Reviewer #3: Yes

Reviewer #5: Yes

6. Review Comments to the Author

Reviewer #3: I believe that with each iteration of revisions, a good text becomes better and better. However, it seemed to me that such a process cannot be continued for too long, because often the better is the enemy of the good.

Reviewer #5: The paper looks much better after the revisions. Almost all my comments are well addressed. I have only a few more comments to be addressed:

> Comparisons DEMATEL with LBWA, FUCOM, BWM and other models is not well presented. More deed discussion about advantages and drawbacks must be presented.

> Source files for LBWA, FUCOM, BWM adn other methods must be cited. Cite references where these methods were presented for the first time.

7. PLOS authors have the option to publish the peer review history of their article (what does this mean?). If published, this will include your full peer review and any attached files.

Reviewer #3: No

Reviewer #5: No

---

## [Author Response · Author response to Decision Letter 3]

14 Nov 2024

Response to Reviewer #3

Comment 1: I believe that with each iteration of revisions, a good text becomes better and better. However, it seemed to me that such a process cannot be continued for too long, because often the better is the enemy of the good.

Response: We appreciate the reviewer’s thoughtful perspective on the iterative refinement process. Indeed, each revision has allowed us to sharpen our analysis and presentation, aligning closely with the high standards of multidisciplinary research studies. While the iterative process is vital for enhancing clarity and precision, we agree with the reviewer’s sentiment that there is a point at which further revisions yield diminishing returns. We have therefore focused this latest revision on addressing the most essential points raised in previous rounds, striving to balance thoroughness with conciseness. We believe the current version effectively communicates our findings while respecting the scope of the study. Thank you for guiding us through this enhancement process.

Response to Reviewer #5

Comment 1: Comparisons DEMATEL with LBWA, FUCOM, BWM and other models is not well presented. More deed discussion about advantages and drawbacks must be presented.

Response: Thank you for highlighting the need for a deeper comparative discussion between DEMATEL and other decision-making models, including LBWA, FUCOM, and BWM. In response, we have expanded our discussion on the comparative strengths and limitations of these methods in complex, multidisciplinary contexts.

While DEMATEL is especially suited to capturing and visualizing causal relationships, particularly in environments where interdependencies and directional influences between factors are prominent, models like LBWA and BWM are typically more efficient for hierarchical decision-making structures. For instance, BWM requires fewer pairwise comparisons, allowing for simpler applications in less interconnected systems. However, it lacks the ability to illustrate the directional causality that is crucial for identifying key influence paths in our study. Similarly, FUCOM and LBWA are effective in cases with distinct, non-recursive criteria relationships, but they do not support dynamic interplays among factors, limiting their applicability in highly interdependent settings.

Our 5-point scale DEMATEL method offers a refined granularity and enhanced precision in influence scoring, allowing us to provide actionable insights by highlighting nuanced dependencies within complex systems. This depth of analysis offers an advantage over the other methods when applied to the intricate environment of multidisciplinary research evaluation. We have added the comparison of the above methods in the revised manuscript, and all the changes are marked with different colors. We hope that our revision can address your concerns.

Comment 2: Source files for LBWA, FUCOM, BWM and other methods must be cited. Cite references where these methods were presented for the first time.

Response: Thank you for the suggestion. We have added references to the foundational works for LBWA, FUCOM, BWM, and related methods to properly acknowledge their original contributions. These citations now appear in the revised manuscript where each method is first discussed, ensuring that readers can trace these methodologies back to their primary sources. All the changes are marked with different colors and we hope that our revision can address your concerns.

---

## [Decision Letter · Decision Letter 4]

25 Nov 2024

Analysis of influencing factors on review efficiency of multidisciplinary scientific research projects using DEMATEL with a 5-point scale

PONE-D-23-29808R4

Dear Dr. Xiao,

We’re pleased to inform you that your manuscript has been judged scientifically suitable for publication and will be formally accepted for publication once it meets all outstanding technical requirements.

Kind regards,

Mehdi Keshavarz-Ghorabaee

Academic Editor

PLOS ONE

Additional Editor Comments (optional):

Reviewers' comments:

Reviewer's Responses to Questions

**Comments to the Author**

1. If the authors have adequately addressed your comments raised in a previous round of review and you feel that this manuscript is now acceptable for publication, you may indicate that here to bypass the “Comments to the Author” section, enter your conflict of interest statement in the “Confidential to Editor” section, and submit your "Accept" recommendation.

Reviewer #5: All comments have been addressed

2. Is the manuscript technically sound, and do the data support the conclusions?

Reviewer #5: Yes

3. Has the statistical analysis been performed appropriately and rigorously? 

Reviewer #5: Yes

4. Have the authors made all data underlying the findings in their manuscript fully available?

Reviewer #5: Yes

5. Is the manuscript presented in an intelligible fashion and written in standard English?

Reviewer #5: Yes

6. Review Comments to the Author

Reviewer #5: The authors have addressed the point of my concern. I am happy with their corrections. Hence, I would like to recommend this manuscript to be published.

7. PLOS authors have the option to publish the peer review history of their article (what does this mean?). If published, this will include your full peer review and any attached files.

Reviewer #5: No

---

## [Editor Report · Acceptance letter]

2 Dec 2024

PONE-D-23-29808R4 

PLOS ONE

Dear Dr. Xiao, 

I'm pleased to inform you that your manuscript has been deemed suitable for publication in PLOS ONE. Congratulations! Your manuscript is now being handed over to our production team.

Kind regards, 

on behalf of

Dr. Mehdi Keshavarz-Ghorabaee 

Academic Editor

PLOS ONE